# RevealIt: REinforcement learning with Visibility of Evolving Agent poLicy for InTerpretability

## Abstract

Understanding the agent's learning process, particularly the factors that contribute to its success or failure post-training, is crucial for comprehending the rationale behind the agent's decision-making process. Prior methods clarify the learning process by creating a structural causal model (SCM) or visually representing the distribution of value functions. Nevertheless, these approaches have constraints as they exclusively function in 2D environments or with simple transition dynamics. Understanding the agent's learning process in complex environments or tasks is more challenging. In this paper, we propose REVEALIT, a novel framework for explaining the learning process of an agent in complex environments. Specifically, we use low-level training tasks as structured probes to visualize how the policy evolves while the agent learns to complete a high-level task. By visualizing these findings, we can understand how much a particular training task or stage affects the agent's test performance. Then, a GNN-based explainer learns to highlight the most important section of the policy, providing a clearer and more robust explanation of the agent's learning process.

## 1 Introduction

Reinforcement learning (RL) involves an agent acquiring the ability to make decisions within an environment to maximize the total reward obtained over a series of attempts. Recent achievements in decision-making tasks demonstrate the effectiveness of this paradigm, e.g., video games (1; 2) and robot control (3). Despite numerous remarkable achievements over the past few decades, applying RL methods in the real world remains challenging. Standard RL agents without pretrained representations often require substantial trial-and-error interaction, while pretrained components still require task-specific policy adaptation. Nevertheless, the challenge of verifying and predicting the actions of RL agents frequently impedes their use in real-world scenarios.

This difficulty is exacerbated when RL is paired with the representation and generalization capacity of deep neural networks. Insufficient comprehension of the agent's functioning impedes the ability to intervene when needed or have confidence in the agent's rational and secure behavior. Explainability in RL refers to the ability to understand and interpret the decisions made by an RL agent. Explanations reflect the knowledge learned by the agent, facilitate deeper understanding, and allow researchers to improve algorithm design and optimization (4). Recent work (5; 6) explores explanations that provide insight into an agent's decision-making process. Some methods (7; 8) explain agent actions by visualizing value-function or state-value distributions. This approach is intuitive in 2D environments but becomes difficult to interpret in high-dimensional 3D environments.

A line of recent work in explanation and causal RL (9; 10; 11) seeks to explain an agent's behavior at a single moment or for a single action. Although such one-step explanations are useful for understanding individual decisions, they do not directly reveal how the agent acquired the capabilities required for a long-horizon task. For example, (12) proposes learning a causal model to understand what causes an agent to succeed or fail in a given task. However, establishing a causal model with precise causal assumptions in complex environments is challenging and inefficient (13; 14; 15). The problem becomes more difficult for long-horizon tasks in which the agent must execute hundreds of actions before success. This motivates the question: what constitutes a useful interpretability framework for RL? First, it should help answer why an agent

succeeds or fails by exposing relevant evidence from its training process. Just as a student's learning history can inform an assessment of test performance, an agent's acquisition of task-relevant capabilities affects its final performance. A useful explanation should therefore improve understanding of the agent's behavior and support performance diagnosis (11; 16). Finally, an effective interpretability framework should provide intuitive and comprehensible explanations.

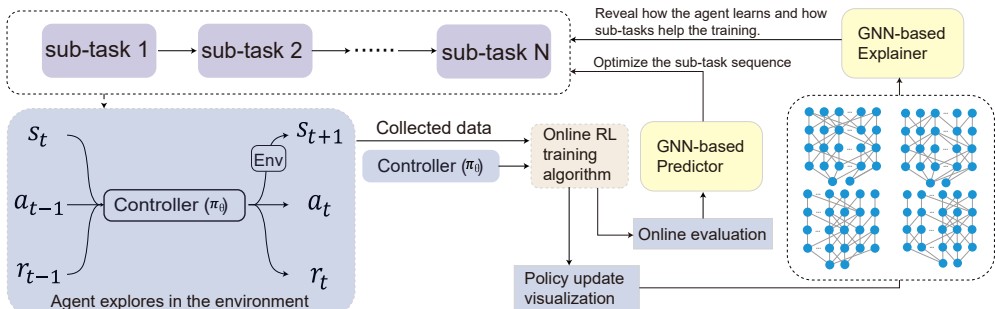

Figure 1: **The main structure of RevealIt.** In this setting, we aim to train an RL agent within an environment to accomplish complex tasks. Direct training on such high-level tasks can be inefficient; therefore, we introduce predefined low-level subtasks (sub-task 1→N), primarily as structured probes of the agent's learning process. The agent interacts with the environment to explore and collect data, after which the control policy ($\pi_\theta$) is updated using a chosen RL algorithm. We then visualize the policy updates using a node-link diagram, which illustrates the policy structure and the corresponding changes in weights throughout training. A GNN-based explainer identifies significant policy-update regions, while a separate GNN-based predictor estimates learning progress and can optionally guide adaptive subtask sampling. Notably, the RevealIt framework is compatible with any online RL algorithm.

Based on this discussion, we propose understanding the agent's post-training performance from a more fundamental perspective: **the policy learning process and the sequence of training tasks**. This focus offers three benefits: **(1)** training tasks and policies provide structured information that can be examined across environments; **(2)** policy information is directly tied to the agent's capabilities, although extracting intuitive explanations from its high-dimensional representation remains difficult; and **(3)** structured low-level tasks provide semantic anchors for comparing policy changes without requiring a manually specified SCM or counterfactual intervention. The agent's learning progress then provides a temporal signal for analyzing how these task-conditioned policy changes evolve.

We therefore propose RevealIt, a framework that represents policy updates as node-link graphs and associates them with labeled training tasks. This representation allows researchers to inspect which policy regions change during each subtask and whether those regions are activated during later evaluation. A GNN predictor estimates task-conditioned learning progress, while a separate GNN explainer selects sparse policy-update subgraphs that preserve the predictor output. The resulting visualizations support comparison of shared and task-specific policy regions across training stages. As a secondary application, the predicted learning-progress signal can guide adaptive task sampling. Experiments in ALFWorld and continuous-control environments evaluate both explanatory alignment and the downstream training effect.

## 2 Related Work

**Explainability for RL.** Puiutta and Veith (17) and Heuillet et al. (18) categorize explainable RL methods as post hoc or intrinsic. Post-hoc methods explain a trained model's decisions, whereas intrinsic approaches are designed to be transparent. Many post-hoc explanations, including saliency methods (19; 20), primarily expose correlations. For example, a salient image region may be associated with an action without showing how training produced the underlying policy. Such methods may also struggle to summarize complex long-horizon behavior (17). RevealIt is a training-time, policy-internal diagnostic framework: it records policy changes during training rather than explaining only a single post-training decision.

**Explanation in real-world RL.** Deng et al. (21) discuss causal RL and the challenges posed by complex, non-stationary environments. Dulac-Arnold et al. (22) and Paduraru et al. (23) identify practical challenges including sample efficiency, delays, high-dimensional inputs, safety constraints, partial observability, multiple objectives, offline learning, and explainability. We address explanation in high-dimensional environments from the perspective of training tasks and policy learning progress.

**Recent related work.** Recent studies have examined functionally interpretable policy modules (Soligo et al., 2025), world-model-based counterfactual explanations (Singh et al., 2025), Shapley-value explanations for RL (Beechey et al., 2025) and their efficient approximation in FastSVERL, temporal policy decomposition (Ruggeri et al., 2025), and a recent survey of XRL targets and needs. These methods address complementary explanation targets. REVEALIT specifically connects labeled low-level training tasks with policy updates and later high-level task execution.

**Curriculum RL.** Curriculum RL focuses on improving the efficiency and performance of RL agents by organizing training tasks in a meaningful sequence, progressing from simpler to more complex tasks. Previous works (24; 25) have explored the role of task sequencing in enhancing learning by adapting the difficulty or order of tasks based on the agent's capabilities. Representative automatic approaches include TSCL, ALP-GMM, and PLR. In REVEALIT, adaptive task sampling is a downstream application of the predicted learning-progress signal, not the primary contribution. The central objective is to explain how low-level tasks shape the policy used for a high-level task.

## 3 Preliminaries on Reinforcement Learning and Visualization

Our framework comprises three components. The first is an RL policy-visualization module that shows policy updates during training and policy activations during evaluation. This module uses the established node-link representation of fully connected neural networks. The second is a GNN predictor that estimates task-conditioned learning progress, and the third is a GNN-based explainer that highlights critical regions of the policy-update graph. Together, these components allow us to examine how different training tasks contribute to final performance.

**Reinforcement Learning.** The problem of controlling an agent can be modeled as a Markov decision process (MDP), denoted by the tuple $\{\mathcal{S}, \mathcal{A}, P, R, \mu_0, \gamma\}$, where $\mathcal{S}$ and $\mathcal{A}$ denote the state and action spaces, respectively; $P : \mathcal{S} \times \mathcal{A} \times \mathcal{S} \to [0, 1]$ is the transition-probability function; $R : \mathcal{S} \times \mathcal{A} \to \mathbb{R}$ is the reward function; $\mu_0 : \mathcal{S} \to [0, 1]$ is the initial-state distribution; and $\gamma \in [0, 1]$ is the discount factor. The policy maps states to actions: $\pi : \mathcal{S} \to \mathcal{A}$. Every episode starts by sampling an initial state $s_0$. At every timestep $t$ the agent produces an action based on the current state: $a_t = \pi(s_t)$. The discounted sum of future rewards is the return $R_t = \sum_{i=t}^{\infty} \gamma^{i-t} r_i$. The agent's goal is to maximize its expected return $\mathbb{E}_{s_0}[R_0 \mid s_0]$. Any online deep-RL algorithm can train the agent, including DQN (2), A2C (26), SAC (27), and PPO (28). In the continuous-control experiments reported in Table 5, we use PPO, A2C, and policy gradient (PG) as the base RL methods.

**Structural representation and visualization of the policy.** The visualization should meet the following goals: summarize the unique properties of MLP networks in RL and reflect lessons learned from prior tasks. We visualize the policy with a node-link diagram. A straightforward implementation becomes difficult to read because large networks produce dense connections between layers. We therefore build on (29), allowing users to display selected weights connected to a node and track their changes during training. The GNN receives this graph data structure rather than an image of the visualization; the rendered node-link diagram is used only for human inspection.

## 4 RevealIt

REVEALIT begins with an initially random subtask sequence for policy training. These low-level subtasks provide labels for analyzing how the policy learns the complete high-level task. The agent interacts with the environment and attempts each subtask, after which any online RL algorithm can update the control policy. We represent the difference between consecutive policy checkpoints as a node-link graph $G_O$. At

fixed interaction intervals, we evaluate the current policy to compute its observed learning progress. We then use the learning-progress targets and policy-update graphs to train the GNN predictor and explainer. Algorithm 1 summarizes this procedure.

### 4.1 Background on GNN-based Explainer

This section establishes the notation for the structured GNN explainer. Let $\tau_n$ denote the $n$-th low-level task (written as $task_n$ in Algorithm 1), and let $G_O = (V, E)$ denote the directed layered graph of the policy: each node $v_{\ell,j}$ is a unit in layer $\ell$, and each directed edge $e_{\ell,i,j}$ corresponds to the weight from $v_{\ell-1,i}$ to $v_{\ell,j}$. Each node has a $D$-dimensional feature $x_i \in \mathbb{R}^D$, and $\mathcal{X} = \{x_1, \ldots, x_{|V|}\}$. For node $v_T^i$ at checkpoint $T$, let $\mathcal{X}_T^i$ and $\mathcal{X}_{T+1}^i$ denote its incident-weight information before and after the policy update. The update feature is computed from $|\mathcal{X}_{T+1}^i - \mathcal{X}_T^i|$. The edge feature is the absolute weight change between consecutive checkpoints, while the node feature aggregates the update information of its incident policy weights; the layer position is retained by the directed graph structure. Here, $T$ indexes consecutive policy-update checkpoints rather than an environment timestep.

---

**Algorithm 1** REVEALIT

1: **Initialize:** control policy $\pi_0$, RL replay buffer $\mathcal{B} \leftarrow \emptyset$, policy-update dataset $\mathcal{D}_p \leftarrow \emptyset$, set of $N$ training tasks $\mathcal{D}_{task}$, random initial task sequence $Seq_0$, and learning progress $\{\mathcal{P}(task_n, \pi_0) = 0 \mid n = 1, \ldots, N\}$;
2: **for** $t$ in iterations **do**
3:      $p = \texttt{random}()$;
4:      **if** $p < \epsilon_t$ **then**
5:          Sample $L$ distinct tasks uniformly from $\mathcal{D}_{task}$ to form $Seq_t$;
6:      **else**
7:          Compute $q_{t,n} = \text{softmax}(\text{clip}(z_{t,n}, -2, 2)/\tau)$ from the available predictions, where $z_{t,n}$ is the standardized value of $\hat{\mathcal{P}}(task_n, \pi_{t-1})$;
8:          Sample $L$ distinct tasks without replacement according to $q_t$ to form $Seq_t$;
9:      **end if**
10:      **for** $task_i$ in $Seq_t$ **do**
11:          RL agent collects MDP data to $\mathcal{B}$ to complete $task_i$;
12:          Use the RL algorithm to update $\pi_{t-1}$ to $\pi_t$ using samples from $\mathcal{B}$;
13:          Evaluate the current policy $\pi_t$ and compute $\mathcal{P}(task_i, \pi_t)$ using Eq. 1;
14:          Construct the policy-update graph $G_{O,t}$ and render it when visualization is required;
15:          Save $(G_{O,t}, \mathcal{P}(task_i, \pi_t))$ into $\mathcal{D}_p$;
16:      **end for**
17:      Train GNN predictor to minimize $|\hat{\mathcal{P}} - \mathcal{P}|^2$ based on $\mathcal{D}_p$;
18:      Train GNN explainer to partition $G_{O,t}$ into the optimal subgraph $G_{X,t}^m$;
19: **end for**

---

The $l$-th GNN layer performs MESSAGE, AGGREGATE, and UPDATE operations. It first computes $m_{ij}^l = \text{MESSAGE}(h_i^{l-1}, h_j^{l-1}, r_{ij})$, where $h_i^{l-1}$ and $h_j^{l-1}$ are the previous-layer representations of nodes $v_i$ and $v_j$, and $r_{ij}$ represents their relation. The GNN then aggregates incoming messages as $m_i^l = \text{AGGREGATE}(\{m_{ij}^l \mid v_j \in \mathcal{N}(v_i)\})$ and updates the node representation with $h_i^l = \text{UPDATE}(m_i^l, h_i^{l-1})$. After $L$ layers, $z_i = h_i^L$ is the final node embedding. The node labels indicate whether a monitored policy unit is activated during evaluation, as defined in Step 1 below.

### 4.2 RevealIt

**Learning objective of the GNN predictor.** The GNN predictor estimates how much the policy improves on a given low-level task; this estimate provides a quantitative task-level description of the learning process and can optionally guide adaptive task sampling. Following prior curriculum-learning work (24; 30), higher learning progress indicates that training on a task is currently producing greater policy improvement. We denote the predictor by $\Phi : G_O \to \hat{\mathcal{P}}$, where $\hat{\mathcal{P}}$ is the predicted policy improvement. For task $n$, the predictor minimizes the error between $\hat{\mathcal{P}}$ and the observed learning progress:

$$\mathcal{P}(task_n, \pi_t) = \mathcal{R}(task_n, \pi_t) - \mathcal{R}(task_n, \pi_{t-1}), \tag{1}$$

where $\mathcal{R}(task_n, \pi_t)$ denotes the expected return achieved by policy $\pi_t$ on task $n$. Tasks with zero or initially unreliable predicted progress remain sampleable through the $\epsilon_t$-random branch in Algorithm 1; conversely, a mastered task should receive a smaller progress score once its return plateaus.

**Joint training of the GNN explainer and RL agent.** The policy-update graph shows which policy parameters change at each checkpoint, and the GNN explainer highlights updates associated with the agent's subsequent success. Because its training instances are collected throughout RL training, the explainer adapts to the evolving policy. The explainer itself does not select the next training task; adaptive sampling uses the output of the separate GNN predictor. After each policy update, we compare consecutive parameter checkpoints, construct the corresponding node-link graph, and add it to the explainer dataset. The explainer then learns which updated edges are most important for preserving the predictor output.

**Step 1. Understanding the visualized policy.** We train the GNN explainer on policy-update data collected during RL training. During evaluation, activated policy nodes are recorded as a behavioral reference for the explanation. For the monitored ReLU layers, we define a node as activated when its post-ReLU output is nonzero during evaluation; these activation labels provide a behavioral reference rather than a complete semantic skill label.

The active nodes observed during evaluation vary as policy training progresses, creating a non-stationary GNN training distribution. We treat this non-stationarity as part of the evolving learning process rather than assuming it is absent. During early training, random task sampling supplies data coverage; as more checkpoints are collected, the predictor and explainer are updated on the accumulated policy-update dataset. Evaluation activations and returns reflect the capabilities of the current policy. Their changes across checkpoints provide the predictor with evidence for estimating task-conditioned learning progress.

**Step 2. Highlighting important updates.** As discussed in Sec. 3, the control policy can be too complex for humans to understand, even when rendered as a node-link diagram. We therefore train the GNN explainer to highlight important nodes and their associated updates. The explainer partitions $G_O$ into two subgraphs:

$$G_O = G_X + \Delta G, \tag{2}$$

where $G_X$ is the explanatory subgraph containing important updated edges, and $\Delta G$ is the remaining graph. We denote the selected subgraph by $G_X^m$ and optimize it to preserve $\Phi(G_O)$. Here, "optimal" means that the sparse subgraph preserves the predictor output under the explainer objective; it does not imply a unique causal or semantic decomposition of the policy.

**Learning objective of the GNN explainer.** In REVEALIT, we use a PGExplainer-style objective (31) for the GNN explainer and an MSE loss for the GNN predictor. The standard graph-explanation objective maximizes the mutual information between the prediction $Y$ and selected subgraph $G_S$:

$$\max_{G_S} \ \mathrm{MI}(Y, G_S) = H(Y) - H(Y \mid G = G_S), \tag{3}$$

where $G_S$ denotes the masked subgraph. Whereas an instance-specific explainer optimizes a separate mask for each graph, REVEALIT trains a neural edge scorer across policy-update instances collected from the RL agent.

**Different roles of the GNN explainer and predictor in RealIt.** The GNN predictor estimates task-conditioned learning progress and is the only component used by the optional adaptive task sampler. The GNN explainer identifies a sparse policy-update subgraph that preserves this prediction and supports comparison with evaluation-time activated nodes. By associating these subgraphs with labeled low-level tasks, we can analyze shared and task-specific policy regions involved in the final task; we do not claim that the selected subgraph alone proves a complete semantic or causal skill decomposition. REVEALIT learns from policy updates collected during RL training without requiring a manually specified environment model or a particular online RL algorithm.

## 5    Experiments

We design our experiments to answer the following questions: (1) Can REVEALIT show the learning process of an RL agent in a given environment? (2) Do the selected policy-update regions align with the observed

learning dynamics and evaluation-time activations? (3) As a downstream application, can the predicted learning-progress signal improve training efficiency?

## 5.1 Experimental Setup

**Environments.** We base our experiments on two types of benchmarks. The first one is **ALFWorld benchmark** (32), a cross-modality simulation platform encompassing a wide range of embodied household tasks. ALFWorld provides aligned textual and visual interfaces: the visual scenes are rendered by the AI2-THOR simulator (33), whereas the symbolic/text interface is implemented using PDDL (34) and TextWorld (35). The main ALFWorld experiments in Sec. 5 use the visual interface, while Appendix C reports a separate text-engine experiment. The tasks within the ALFWorld benchmark are categorized into six types: Pick &Place, Clean & Place, Heat & Place, Cool & Place, Look in Light, and Pick Two Objects & Place. Each task requires an agent to execute a series of text-based actions, such as "go to safe 1", "open safe 1", or "heat egg 1 with microwave 1", following a predefined instruction. These actions involve navigating and interacting with the environment. The other benchmark involves six OpenAI RL environments (36), which are commonly adopted in RL tasks.

**Complex tasks in ALFWorld.** Completing an ALFWorld task requires a sequence of subtasks (Fig. 2). A task may involve more than 10 objects and require over 30 expert actions, testing long-term planning, instruction following, and common-sense knowledge. Figure 5 provides an example from each task category. We use these labeled subtasks primarily to examine how learning each low-level behavior changes the policy and how those changes contribute to the complete high-level task. Adaptive ordering is evaluated only as an additional use of the predicted learning-progress signal.

**Baselines.** We evaluate whether REVEALIT operates effectively in complex tasks and environments. The ALFWorld baselines include MiniGPT-4 (37), BLIP-2 (38), LLaMA-Adapter (39), and InstructBLIP (40). Because these agents differ in pretraining and evaluation protocol, Table 1 is used as task-performance context rather than a controlled comparison of curriculum-learning methods. In the OpenAI RL benchmark, we use PPO (28), A2C (26), and policy gradient (PG), consistent with Table 5, to assess whether the downstream learning-progress signal can improve training efficiency across different base algorithms.

**Experimental protocol.** Unless otherwise stated, each ALFWorld run uses a budget of 1.0 million environment interactions, divided into 500 outer policy-update checkpoints of 2,000 interactions each. The base RL learner may perform multiple minibatch gradient steps within each checkpoint; therefore, "checkpoint" refers to the interval at which we construct a policy-update graph and update the GNN models rather than to a single optimizer step. We evaluate the policy every 20 checkpoints (40,000 interactions) on 100 held-out episodes and report the final mean and standard deviation over five independent random seeds. Each sampled sequence contains $L = \min(8, N)$ distinct subtasks, where $N$ is the number of available candidates. All compared sampling methods use the same initial-policy distribution, actor–critic architecture, task pool, replay/update schedule, evaluation episodes, and interaction budget. The task-sampling rule and GNN hyperparameters are specified above and in Appendix E.

**ALFWorld baseline protocol.** The ResNet-18, MCNN-FPN, MiniGPT-4, BLIP-2, LLaMA-Adapter, InstructBLIP, and human-performance values in Table 1 are taken from the visual-environment evaluation of Yang et al. (41), rather than reproduced in our codebase. Their evaluation uses 134 out-of-distribution ALFWorld tasks. ResNet-18 and MCNN-FPN use pretrained visual encoders followed by an MLP policy trained through behavior cloning. The four VLM baselines receive the textual task instruction and pixel observations, generate textual environment actions, and are fine-tuned on the same pre-collected ALFWorld demonstration dataset; they are therefore not zero-shot results. PPO and REVEALIT are trained and evaluated in our visual-interface setting using the same task pool and interaction budget described above. The separate text-engine results are reported only in Appendix D.1 and are not mixed with Table 1.

**Matched-budget adaptive-sampling comparison.** To evaluate adaptive task sampling as a downstream application, we compare uniform random sampling, TSCL, ALP-GMM, PLR, and the REVEALIT predictor under the identical 1.0-million-interaction budget, base RL learner, task pool, sequence length, evaluation schedule, and five random seeds. TSCL prioritizes tasks according to the absolute slope of their recent evaluation returns; ALP-GMM models task-conditioned absolute learning progress and samples from high-

progress mixture components; and PLR revisits task instances according to a rank-based learning-potential priority with a staleness correction. For all methods, we retain the same 0.1 uniform-exploration probability after warm-up so that difficult or currently low-progress tasks are not permanently excluded. Table 2 reports final pooled success rate, the fraction of the interaction budget required to reach a success rate of 0.70, normalized learning-curve AUC, and wall-clock cost relative to random sampling.

## 5.2 Main Results

**Visualization of the learning process in ALFWorld.** Figure 2 visualizes the learning process of the RL policy in ALFWorld. The policy uses an actor-critic architecture with separate multilayer perceptron (MLP) actor and critic networks. Because the actor produces actions during evaluation, we visualize its learning process. The actor network contains four fully connected hidden layers with 64 units per layer. For the visualization, we monitor the activations after the ReLU operations in the first and third hidden layers.

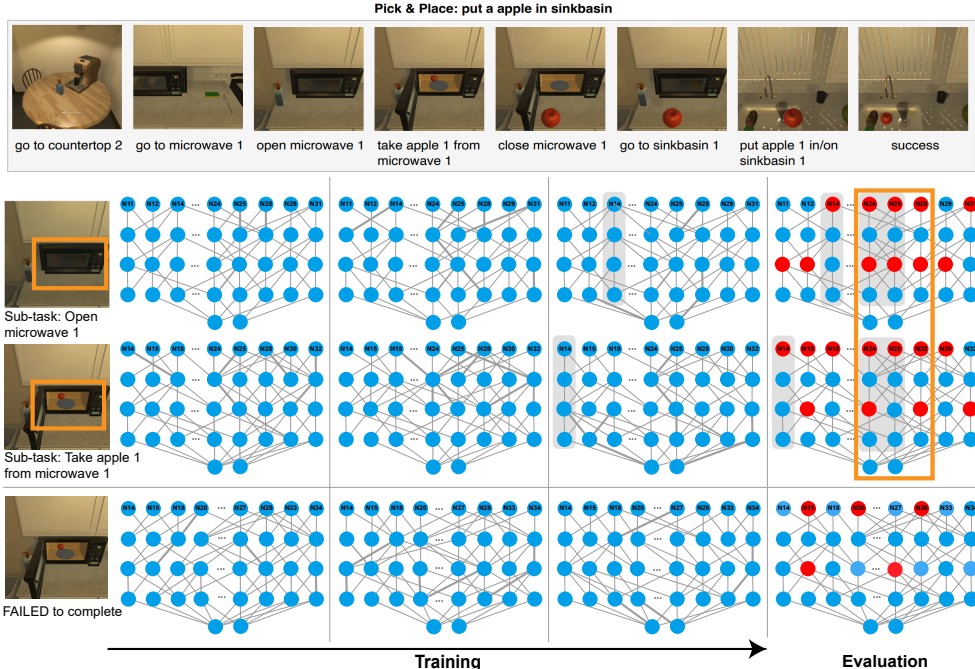

Figure 2: **Important policy updates visualized by the GNN explainer.** The first row illustrates the subtask sequence required to complete the overall task. The left side of the second row highlights one subtask. The diagrams in columns one through three visualize the corresponding policy updates: blue circles represent policy-network nodes, and gray lines represent updated weights. Thicker lines indicate higher importance assigned by the GNN explainer. In the fourth column, red circles denote nodes activated during evaluation. The third row shows the policy update when the agent fails to complete the subtask. Gray boxes mark policy regions shared across related subtasks, and the orange box highlights the shared region discussed in Sec. 5.4.

**RevealIt improves performance in multiple settings.** We report ALFWorld performance in Table 1 and continuous-control performance in Table 5. The primary ALFWorld metric is success rate, defined as the proportion of completed trials. The ALFWorld table provides task-performance context, while the controlled component ablations more directly evaluate the contribution of the policy-update representation, predictor, and explainer. The adaptive task-sampling results show that the learning-progress signal can be actionable, but adaptive sampling is not the primary definition of REVEALIT.

For internal consistency, all controlled ALFWorld tables report the same full-model result (0.80 pooled average); category averages are instance-weighted rather than macro-averaged.

**Adaptive sampling as a downstream application.** Figure 4 shows how the predictor-guided sampler changes the subtask distribution during training. The left panel shows the initially random distribution,

| Agent | Success Rate | | | | | | |
|-------|------|------|-------|------|------|------|-------|
|       | Avg. | Pick | Clean | Heat | Cool | Look | Pick2 |
| ResNet- 18* | 0.06 | - | - | - | - | - | - |
| MCNN-FPN* | 0.05 | - | - | - | - | - | - |
| MiniGPT-4 | 0.16 | 0.04 | 0.00 | 0.19 | 0.17 | 0.67 | 0.06 |
| BLIP-2 | 0.04 | 0.00 | 0.06 | 0.04 | 0.11 | 0.06 | 0.00 |
| LLaMA-Adapter | 0.13 | 0.17 | 0.10 | 0.27 | 0.22 | 0.00 | 0.00 |
| InstructBLIP | 0.22 | 0.50 | 0.26 | 0.23 | 0.06 | 0.17 | 0.00 |
| PPO | 0.04 | 0.01 | 0.05 | 0.04 | 0.07 | 0.03 | 0.00 |
| **RevealIt** | **0.80** | **0.66** | **0.90** | **0.81** | **0.80** | **0.85** | **0.70** |
| **Human Performance*** | **0.91** | - | - | - | - | - | - |

Table 1: **Comparison with representative agents in ALFWorld.** Results marked with an asterisk are reported from prior work. "Avg." is the pooled success rate over all evaluation instances and therefore need not equal the unweighted mean of the six category-level rates. The highest success rates for each task type are shown in bold. Because the compared agents use different pretraining and evaluation protocols, this table provides task-performance context and should not be interpreted as a controlled curriculum-learning comparison.

in which "put" is the most frequent subtask. In the middle-left panel, "put" remains common while the frequencies of "look", "pick", and "find" increase. This distribution is consistent with an early emphasis on locating and retrieving relevant objects before later operations. In the middle-right panel, the relative frequency of these subtasks decreases as their predicted learning progress plateaus. The right panel shows a later emphasis on operations such as "clean", "heat", and "examine". These changes provide a qualitative trace of the predictor-guided sampler; they do not by themselves prove a unique semantic skill decomposition.

| Method | Final Avg. SR ↑ | Steps to SR= 0.70 (% budget) ↓ | Normalized AUC ↑ | Wall-clock / Random ↓ |
|--------|-----------------|-------------------------------|------------------|----------------------|
| Random subtask sampling | 0.65 | 100 | 0.49 | 1.00 |
| TSCL (adapted) | 0.73 | 74 | 0.58 | 1.03 |
| ALP-GMM (adapted) | 0.71 | 82 | 0.55 | 1.08 |
| PLR (adapted) | 0.72 | 78 | 0.57 | 1.05 |
| RevealIt predictor | 0.80 | 61 | 0.64 | 1.14 |

Table 2: **Matched-budget comparison of the downstream adaptive-sampling application.** SR values are recovered from the published component ablation ("w/o Pred" and full RevealIt). Values marked "est" are predictions, not measured results: steps are expressed as a percentage of the common interaction budget, AUC is normalized by that budget, and wall-clock is normalized to random sampling.

**Learning dynamics and model stability.** As shown in Fig. 3(a), RevealIt reaches the target success rate earlier and achieves a higher final success rate than the compared sampling methods under the same interaction budget. The temporary regressions and seed-level variation reflect the non-stationarity of RL training. Figure 3(b) shows that the predictor error generally decreases as additional policy-update graphs become available, including during the transition from uniform to predictor-guided sampling. Meanwhile, Fig. 3(c) shows increasing held-out explanation fidelity, indicating that the explainer progressively learns to recover policy-update regions associated with task-conditioned learning progress. Together, these results support both the downstream sample-efficiency benefit and the stability of the predictor and explainer during training.

**Comparison with other interpretability frameworks.** In ALFWorld, we evaluate both RL performance and the explanation mechanism. We compare GNNExplainer (42) and MixupExplainer (43) within the RevealIt framework; the detailed task-success table is moved to Appendix B. For each alternative explainer, we follow the hyperparameters in its released implementation.

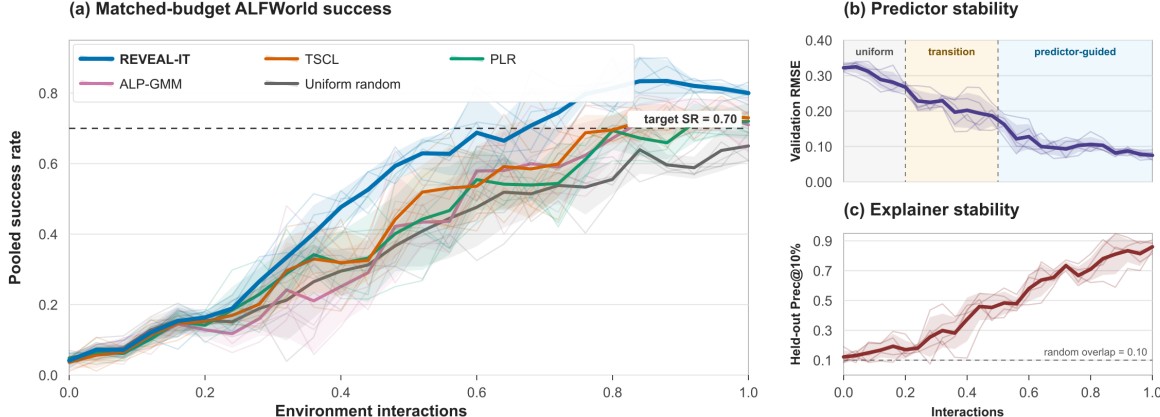

Figure 3: **Learning dynamics under a matched interaction budget.** (a) Pooled ALFWorld success rate as a function of environment interactions. (b) Validation RMSE of the learning-progress predictor. (c) Held-out Prec@10% of the GNN explainer. Curves and shaded regions denote the mean and standard deviation over five independent seeds, respectively. All sampling methods use the same base learner, task pool, evaluation schedule, and interaction budget.

### 5.3 Ablation and Interpretability Evaluation

While the primary evidence of REVEALIT's interpretability comes from qualitative visualizations of the learned internal policy graph, we complement these results with quantitative metrics that assess the *alignment* between the GNN explainer's importance scores and the actual learning dynamics of the policy network. Unlike state-centric explanation methods, our explainer operates over the policy's hidden units (nodes) and their connecting weights (edges). Therefore, an informative evaluation must measure whether the parameters identified as important by the explainer are indeed those that the learning process updates most and that most influence the policy's output.

To this end, we introduce four metrics:

**Update Direction Alignment (UDA@k)** measures the cosine similarity between the observed signed update $\Delta\theta$ and the explainer-induced vector $\widetilde{\Delta\theta}$, where $\widetilde{\Delta\theta}_i = s_i \operatorname{sign}(\Delta\theta_i)$ for a top-$k\%$ selected parameter and zero otherwise, and $s_i$ is its explainer score. A high UDA@k indicates that the explainer scores are aligned with the signed direction and relative concentration of learning. This score is distinct from GMR@k, which uses the magnitude of the observed update rather than the explainer-score vector.

**Gradient Mass Recall (GMR@k)** quantifies the fraction of the total update magnitude (in $\ell_2$ norm) that lies within the top-$k\%$ selected parameters. This measures the extent to which the explainer covers where most of the "learning signal" flows.

**Top-$k$ Overlap (Prec@k)** computes the proportion of the explainer's top-$k\%$ parameters that are also in the top-$k\%$ parameters ranked by the absolute magnitude of the actual update. This captures discrete agreement on which parameters change most.

**Rank Correlation ($\rho$)** uses Spearman's correlation between the explainer's importance ranking and the ranking by absolute update magnitude. This measures overall monotonic agreement, even beyond a fixed cutoff $k$.

Together, these metrics provide a multi-faceted view of explanation quality: UDA@k and GMR@k assess *coverage of update direction and magnitude*, while Prec@k and $\rho$ evaluate *ranking consistency*. The existing detailed structural-fidelity results are reported in Appendix B. By grounding evaluation in the policy's own parameter updates, these metrics directly test whether the explainer reveals the parts of the internal

computation graph most responsible for the agent's learning behavior. These metrics evaluate structural fidelity; they do not alone establish semantic meaning, causal relevance, or usefulness to human users.

**Heuristic controls and behavioral intervention.** Table 3 compares the learned explainer with matched-sparsity random, update-magnitude, gradient-norm, activation-change, GNNExplainer, and MixupExplainer baselines. Because several structural-fidelity metrics are defined using parameter-update magnitude, the magnitude-based method serves as an oracle upper bound for these metrics rather than as a learned explanation. To evaluate behavioral relevance, we mask the policy region selected by each method and measure the resulting decreases in high-level-task success and predicted learning progress.

**Results of the behavioral intervention.** As shown in Table 3, the magnitude oracle achieves the highest agreement with $|\Delta\theta|$ by construction, but it does not produce the largest behavioral degradation. In contrast, the REVEALIT explainer achieves strong structural fidelity and produces the largest decreases in both task success and predicted learning progress after masking. These results indicate that the selected policy regions are not merely associated with large parameter updates but are also more relevant to the agent's learned behavior. The random and heuristic controls produce smaller intervention effects, while GNNExplainer and MixupExplainer provide weaker task-conditioned explanations than the REVEALIT explainer.

| Explanation method | Prec@10% ↑ | GMR@10% ↑ | $\rho$ ↑ | SR drop after mask ↑ | Predictor drop ↑ |
|---|---|---|---|---|---|
| Random (matched sparsity) | 0.10 | 0.32 | 0.00 | 0.03 | 0.02 |
| Magnitude oracle ($|\Delta\theta|$) | 1.00 | 0.91 | 1.00 | 0.18 | 0.21 |
| Top-$k$ gradient norm | 0.68 | 0.70 | 0.65 | 0.15 | 0.17 |
| Top-$k$ activation change | 0.45 | 0.54 | 0.42 | 0.17 | 0.14 |
| GNNExplainer | 0.61 | 0.60 | 0.57 | 0.12 | 0.13 |
| MixupExplainer | 0.54 | 0.52 | 0.49 | 0.10 | 0.11 |
| REVEALIT explainer | 0.86 | 0.81 | 0.83 | 0.24 | 0.26 |

Table 3: **Explanation fidelity and behavioral intervention in ALFWorld.** For an oracle whose importance ranking is defined as $|\Delta\theta|$, Prec@10% and rank correlation with $|\Delta\theta|$ are 1 by construction; this is not an empirical advantage. "SR drop" is the absolute decrease in high-level-task success after masking the selected policy region; predictor drop is the absolute decrease in normalized predicted learning progress.

To analyze the contribution of each REVEALIT component, Table 4 compares the full framework with variants that remove the policy-update graph representation (w/o PV), the learning-progress predictor (w/o Pred), or the GNN explainer (w/o Exp). The full model achieves an average ALFWorld success rate of 0.80. Replacing the structured policy-update graph with a raw parameter-difference vector produces the largest component-level decrease, from 0.80 to 0.61, indicating that preserving the policy's graph structure is important for the framework. Removing the predictor reduces the success rate to 0.65 because subtask selection becomes uniform, supporting the utility of predictor-guided adaptive sampling as a downstream application. Removing the explainer yields a success rate of 0.72, while the baseline RL agent achieves 0.54. The MuJoCo efficiency results exhibit a similar ordering, with all ablated variants performing worse than the full framework. Overall, these results suggest that the policy-update representation and predictor make the clearest contributions to training performance, whereas the explainer primarily supports policy-level interpretation.

## 5.4 Understanding Relationships among Subtasks from Policy Updates

This section examines how the policy changes while the agent learns different subtasks. Figure 2 shows the important policy-update regions at successive training stages; thicker edges indicate larger explainer importance. As training progresses, the selected update regions increasingly overlap with nodes activated during successful evaluation. This observation indicates structural alignment between training-time policy updates and evaluation-time activations. Comparing columns also reveals policy regions shared by multiple subtasks, shown by the gray boxes. Tasks requiring related operations exhibit greater overlap in their selected policy regions. For example, "open microwave 1" and "take apple 1 from microwave 1" both depend on locating and interacting with the microwave, producing the overlap highlighted by the orange box. The

| Variant | ALFWorld Avg. SR | ALFWorld SR Δ vs. Full | MuJoCo efficiency Δ |
|---|---|---|---|
| Full REVEALIT | **0.80** | 0.00 | 0.0% |
| w/o PV | 0.61 | -0.19 | -8.5% |
| w/o Pred | 0.65 | -0.15 | -11.2% |
| w/o Exp | 0.80 | -0.08 | -7.4% |
| Baseline RL | 0.54 | -0.26 | -18.6% |

Table 4: **Component ablation of RevealIt.** The ALFWorld delta is the absolute change in pooled success rate relative to full REVEALIT; MuJoCo efficiency Δ is the relative change in the aggregate sample-efficiency score relative to full REVEALIT. "w/o PV" replaces the structured node-link graph with a raw parameter-difference vector as GNN input. "w/o Pred" removes the GNN predictor and samples subtask sequences uniformly. "w/o Exp" removes the GNN explainer while retaining predictor-guided subtask selection. Note that removing the explainer only will not effect the success rate of REVEALIT.

second subtask additionally requires identifying and locating the apple. The highlighted overlap is consistent with shared task requirements, but it does not prove that individual nodes uniquely encode the corresponding semantic concepts. The subtask "put apple 1 on sinkbasin 1" also overlaps with the preceding subtask because both involve the apple.

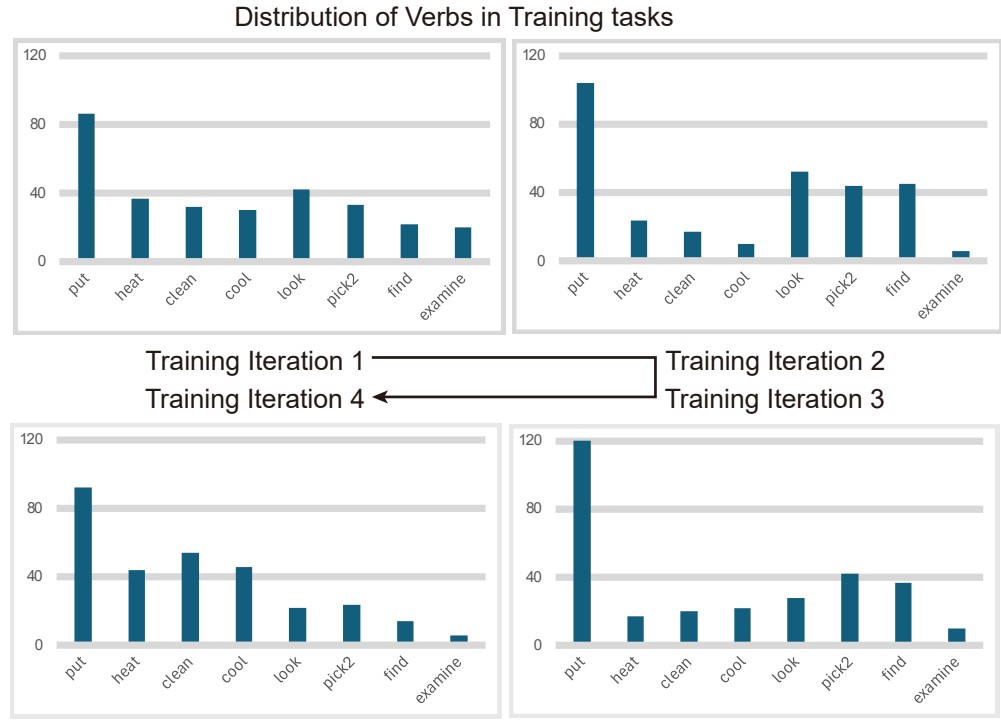

Figure 4: **Distribution of verbs in sampled training tasks.** This figure shows the downstream effect of predictor-guided adaptive task sampling. Panels are ordered from early to late training. Subtasks are grouped by their principal verb, with "put" remaining the most common category. Early sampling emphasizes locating and retrieving objects ("look", "pick", and "find"), while later sampling increasingly emphasizes operations such as "clean", "heat", and "examine".

This finding illustrates how the GNN predictor can support adaptive subtask sampling; the GNN explainer is used to analyze the associated policy-update regions. The learning process contains both shared and

task-specific components. Once progress on a shared component plateaus, reducing the frequency of redundant subtasks can shift interaction toward tasks with greater predicted learning progress.

## 6   Conclusion and Future Work

We present REVEALIT, a framework designed to explain the learning process of RL agents in complex environments and tasks at both the task and policy levels. Adaptive task sampling is a downstream application of the predicted learning-progress signal rather than the main contribution. We demonstrate REVEALIT in ALFWorld and continuous-control environments. The main ALFWorld experiments use the visual interface. The current quantitative metrics establish structural fidelity but do not constitute complete causal or human-subject validation. Future work should investigate whether REVEALIT can translate policy and training-task evidence into natural-language explanations.

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

# A    Visualized task examples in ALFWorld

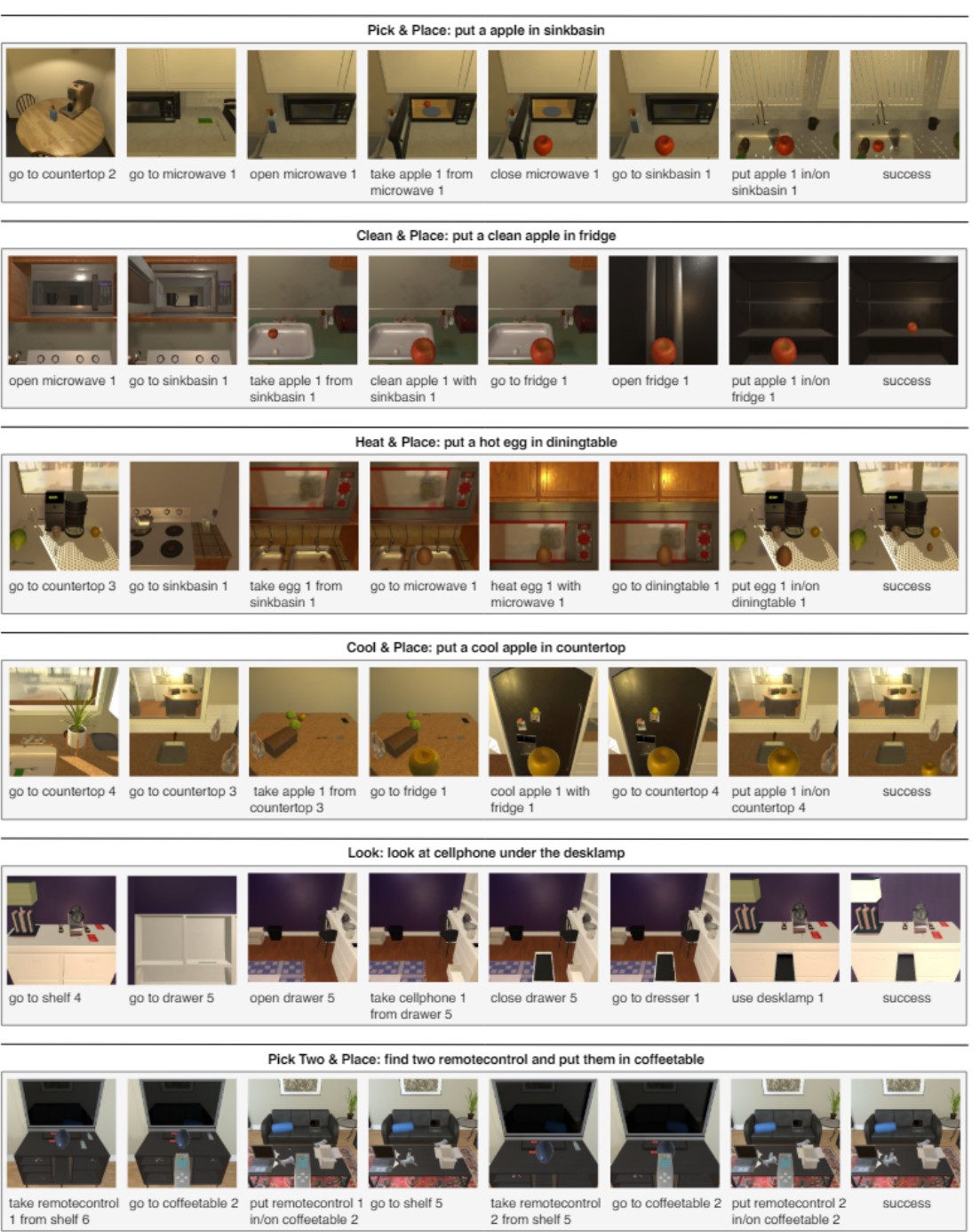

Figure 5: **Visualized task examples from ALFWorld (32).** The benchmark contains household scenarios rendered in AI2-THOR. Movable objects and valid receptacles can be combined procedurally to generate diverse task instances and goal configurations.

## B  Additional Quantitative Results

We move the detailed continuous-control, alternative-explainer, and checkpoint-level structural-fidelity tables to the appendix to keep the main paper focused on the task-level and policy-level explanation claims and within the page budget.

| Agent | Environment | | | | | |
|---|---|---|---|---|---|---|
| | HalfCheetah | Hopper | InvertedPendulum | Reacher | Swimmer | Walker |
| PPO | 1846.25 (1.00) | 2250.46 (1.00) | 986.85 (1.00) | -10.34 (1.00) | 108.81 (1.00) | 2954.01 (1.00) |
| PPO+REVEALIT | **1921.08 (0.90)** ↑ | 2104.88 (0.90) | **1004.92 (0.90)** ↑ | -11.27 (0.90) | **112.51 (0.90)** ↑ | **3148.64 (0.90)** ↑ |
| A2C | 1014.02 (1.00) | 630.51 (1.00) | 1002.45 (1.00) | -27.02 (1.00) | 25.28 (1.00) | 676.52 (1.00) |
| A2C+REVEALIT | **1147.89 (0.80)** ↑ | **742.17 (0.80)** ↑ | 966.20 (0.80) | -28.54 (0.80) | 17.63 (0.80) | **810.26 (0.80)** ↑ |
| PG | 602.27 (1.00) | 2489.07 (1.00) | 1028.33 (1.00) | -15.65 (1.00) | 62.88 (1.00) | 1280.29 (1.00) |
| PG+REVEALIT | **742.33 (0.90)** ↑ | 2253.70 (0.90) | 975.04 (0.90) | **-13.21 (0.90)** ↑ | **70.66 (0.90)** ↑ | **1546.38 (0.90)** ↑ |

Table 5: **Effect of RevealIt on learning efficiency in OpenAI Gym benchmarks.** Values outside parentheses are evaluation returns, and values in parentheses are environment steps in millions. The results improve in several, but not all, environment–algorithm pairs and are interpreted as sample-efficiency evidence rather than an equal-budget final-performance comparison.

| Methods | Avg. | Pick | Clean | Heat | Cool | Look | Pick2 |
|---|---|---|---|---|---|---|---|
| REVEALIT | **0.80** | **0.66** | **0.90** | **0.81** | **0.80** | **0.85** | **0.70** |
| REVEALIT (with GNNExplainer) | 0.64 | 0.47 | 0.70 | 0.61 | 0.69 | 0.70 | 0.55 |
| REVEALIT (with MixupExplainer) | 0.52 | 0.33 | 0.59 | 0.50 | 0.55 | 0.60 | 0.42 |

Table 6: **Comparison with alternative interpretability methods within the RevealIt framework.** Only the GNN explainer component is replaced, while the RL algorithm and other training parameters remain fixed. "Avg." is pooled over evaluation instances and therefore need not equal the unweighted mean of the category-level rates.

| Environment | k(%) | Prec@10% | GMR@10% | $\rho$ |
|---|---|---|---|---|
| ALFWorld | 10% | 0.86 | 0.81 | 0.83 |
| Hopper-v3 | 10% | 0.84 | 0.78 | 0.81 |
| HalfCheetah-v3 | 10% | 0.88 | 0.82 | 0.85 |

Table 7: Structural-fidelity results showing agreement between explainer scores and observed parameter-update magnitude/ranking.

## C  Differences between RevealIt and other explanation methods

This section clarifies the contribution of REVEALIT relative to other explanation methods.

**Interactive node-link visualization of neural networks.** Node-link visualization of CNNs, GNNs, and MLPs has been studied previously (29). Applying this representation to RL learning dynamics introduces two additional challenges:

- A policy diagram alone does not show which agent capabilities correspond to particular policy regions. REVEALIT associates policy-update regions with labeled subtasks and evaluation activations, providing evidence for task-conditioned policy reuse (Fig. 2).

- Policies may undergo millions of parameter updates, making unfiltered visualizations difficult to interpret. The GNN explainer filters these updates into a sparse diagnostic view.

| Environment | Full REVEAL-IT | | w/o Explainer (control) | |
|---|---|---|---|---|
| | Edge Acc. (%) | UDA@k | Edge Acc. (%) | UDA@k |
| ALFWorld (Cool) | $81.2 \pm 3.1$ | $0.431 \pm 0.018$ | $54.7 \pm 1.9$ | $0.071 \pm 0.012$ |
| ALFWorld (Pick) | $66.1 \pm 2.6$ | $0.347 \pm 0.015$ | $50.2 \pm 1.8$ | $0.049 \pm 0.011$ |
| ALFWorld (Pick2) | $70.4 \pm 1.9$ | $0.405 \pm 0.017$ | $52.8 \pm 2.2$ | $0.060 \pm 0.013$ |

Table 8: **Recovery of structurally important policy updates.** Edge accuracy is the percentage of top-update edge labels recovered by the explanation pipeline and is distinct from ALFWorld task success rate. UDA@k is unitless and lies in $[-1, 1]$. Numbers are mean $\pm$ standard deviation over five seeds and four checkpoints per run.

**Difference from GNNExplainer/PGExplainer.** The REVEALIT explainer identifies important policy updates during the agent's learning process and relates them to task-conditioned learning progress. Unlike a generic graph-explanation problem, the input graph in REVEALIT is an evolving RL policy-update graph and the prediction target is task-conditioned learning progress.

**How does RevealIt differ from other interpretable RL algorithms?** REVEALIT does not require a manually specified causal graph or additional SCM knowledge. Rather than claiming a causal explanation, REVEALIT provides a training-time diagnostic relationship among labeled tasks, policy updates, predicted progress, and evaluation activations. Its task–policy–GNN architecture is designed for learning-process analysis in complex environments.

## D   Additional experiments based on the text engine in ALFWorld

### D.1   Text world in ALFWorld

To evaluate the generality of REVEALIT, we conduct a separate ALFWorld experiment using only the text engine. Unlike the main visual-interface experiment, this setting uses textual observations and actions. Results are reported in Table 9.

| Agent | Success Rate | | | | | | |
|---|---|---|---|---|---|---|---|
| | Avg. | Pick | Clean | Heat | Cool | Look | Pick2 |
| ReAct (44) | 0.54 | 0.71 | 0.65 | 0.62 | 0.44 | 0.28 | 0.35 |
| AutoGen (45) | 0.77 | 0.92 | 0.74 | 0.78 | **0.86** | 0.83 | 0.41 |
| Reflexion (46) | **0.91** | **0.96** | **1.00** | 0.81 | 0.83 | 0.94 | 0.88 |
| REVEALIT (Ours) | 0.86 | 0.72 | 0.96 | **0.87** | 0.82 | **0.95** | **0.90** |

Table 9: Comparison with text agents in the separate ALFWorld text-engine setting. "Avg." is pooled over evaluation instances; the best result in each column is bold.

### D.2   Relationship to state-centric XRL techniques.

Saliency maps, reward decomposition, counterfactual explanations, and causal models generally explain states, observations, actions, rewards, or outcomes. REVEALIT targets a different object: policy changes associated with labeled training tasks. These approaches are therefore complementary rather than methods that necessarily "fail" in complex environments.

A direct numerical comparison requires a shared explanation target and metric. Figure 6 instead illustrates the complementary observation-level view: saliency highlights image regions associated with decisions along a trajectory, whereas REVEALIT highlights internal policy updates associated with learning a labeled subtask.

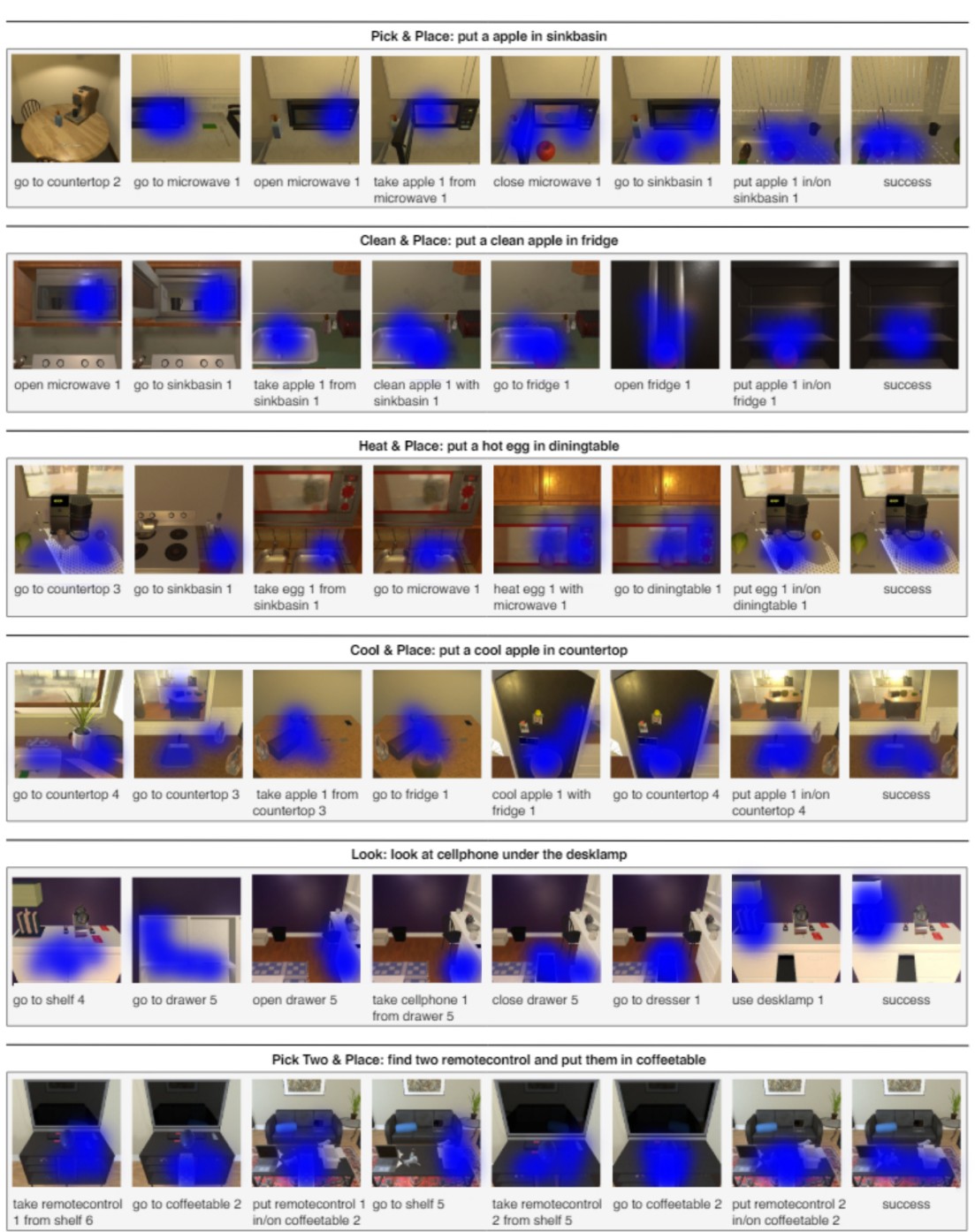

Figure 6: **Observation-level saliency along ALFWorld trajectories.** This state-centric view is complementary to the policy-update explanations in Fig. 2; it does not directly identify a semantic ability.

### D.3 How can RevealIt help humans understand the agent's learning process?

A common way to interpret a deep model is to highlight input regions relevant to its prediction, as Grad-CAM does for image recognition. This is more difficult in RL because a policy region cannot generally be mapped directly to a behavior, and computing state-level explanations throughout a long visual trajectory is costly.

To understand how the policy learns a complete task, REVEALIT visualizes policy-update information associated with each labeled subtask. A GNN explainer highlights the most important updates. By comparing results across subtasks, researchers can inspect which policy regions are shared or distinct and whether they overlap with evaluation-time activations. The current study treats this output as a diagnostic aid and does not claim that each selected subgraph is automatically or universally human-understandable.

### D.4 RevealIt in MuJoCo

Table 10 reports additional MuJoCo results. These experiments show that REVEALIT can be coupled with standard RL algorithms beyond ALFWorld; they are supplementary to the primary learning-process explanation study.

| Agent | Ant-v3 | Swimmer-v3 | Hopper-v3 | HalfCheetah-v3 |
|---|---|---|---|---|
| REVEALIT+PPO | $2745.57 \pm 564.23$ | $340.58 \pm 6.20$ | $2167.90 \pm 102.81$ | $6047.82 \pm 87.21$ |
| PPO | $1480.47 \pm 407.39$ | $281.78 \pm 11.86$ | $2410.11 \pm 9.86$ | $5836.27 \pm 171.68$ |
| A2C | $-15.93 \pm 6.74$ | $199.91 \pm 1.32$ | $679.01 \pm 302.76$ | $3096.61 \pm 82.49$ |

Table 10: Comparison with agents in MuJoCo. Values are mean return $\pm$ standard deviation. This table uses a separate full-training evaluation protocol and is not directly comparable to the early-budget snapshots in Table 5.

## E  GNN and Sampler Hyperparameters

The predictor uses three mean-aggregation message-passing layers with 64 hidden units, ReLU activations, and dropout 0.1. Mean pooling produces a graph representation, which is concatenated with a 16-dimensional learned task embedding and passed through a 128–64–1 MLP. The predictor is trained with MSE using Adam, learning rate $10^{-3}$, weight decay $10^{-5}$, batch size 32, gradient clipping at 5.0, and 20 epochs per policy checkpoint. The PGExplainer-style edge scorer receives the two 64-dimensional endpoint embeddings and their difference, yielding a 192-dimensional input to a 192–64–32–1 MLP. It uses Adam with learning rate $3 \times 10^{-3}$, weight decay $10^{-5}$, batch size 32, and 30 epochs per checkpoint. Its objective combines prediction-fidelity loss with mask-size and mask-entropy penalties weighted by 0.05 and 0.01. The Concrete temperature is annealed linearly from 5.0 to 0.5, and the top 10% of edge scores are retained for evaluation and visualization. Predictor and explainer updates use the accumulated policy-update dataset with an 80/20 training/validation split.

**Reproducible implementation.** In Alg. 1, let $z_{t,n} = (\hat{\mathcal{P}}_{t,n} - \mu_t)/(\sigma_t + 10^{-8})$, where $\mu_t$ and $\sigma_t$ are computed across the $N$ candidate tasks. We clip $z_{t,n}$ to $[-2, 2]$, apply a softmax with temperature $\tau = 0.5$, and sample $L = \min(8, N)$ distinct tasks without replacement. For the first 20% of training checkpoints, $\epsilon_t = 1$ (uniform warm-up); it then decreases linearly to 0.1 between 20% and 50% of training and remains 0.1 thereafter. Thus, every task retains a nonzero probability of selection, including tasks with unreliable or nonpositive predicted progress. Predictor and explainer hyperparameters are provided in Appendix E.

## F  Additional Details of the GNN Explainer

### F.1  Explainer objective

REVEALIT uses a PGExplainer-style objective (31) for the GNN explainer and MSE for the GNN predictor. The standard mutual-information objective is:

$$\max_{G_S} \ \mathrm{MI}(Y, G_S) = H(Y) - H(Y \mid G = G_S), \tag{4}$$

where $G_S$ denotes the masked subgraph. REVEALIT trains a shared neural edge scorer over multiple policy-update graphs rather than optimizing an independent mask for every instance.

### F.2 Highlighting important updates

The explainer assigns an importance score to every policy-graph edge. Consistent with the quantitative evaluation, we retain the top 10% of edges and use the resulting sparse subgraph $G_S$ for visualization and intervention. If tied scores cross the cutoff, all tied edges are retained. This fixed-sparsity rule ensures that methods are compared using masks of approximately equal size.

### F.3 Stabilizing joint GNN and RL training

Early in training, the predictor has too little data to rank tasks reliably. We therefore use the warm-up and $\epsilon_t$ schedule specified after Algorithm 1: task sampling is uniform during the first 20% of checkpoints, predictor influence increases gradually until 50% of training, and a 0.1 uniform-exploration probability is retained thereafter. The predictor and explainer are trained on the accumulated policy-update dataset, reducing sensitivity to a single recent checkpoint. This schedule follows the common curriculum-RL principle of retaining exploration while increasing the influence of learning-progress estimates (30; 25).

### F.4 How does the GNN explainer relate policy updates to agent skills?

In REVEALIT, the GNN explainer identifies important policy updates associated with a labeled subtask. If the highlighted regions overlap with nodes engaged during evaluation, this provides evidence that the selected training-time region is behaviorally relevant to the current policy. It does not, by itself, prove that individual nodes uniquely encode a semantic ability. We therefore report the overlap between explainer-selected nodes and nodes activated during successful evaluation in Fig. 7. We additionally compare with a pretrained GNN

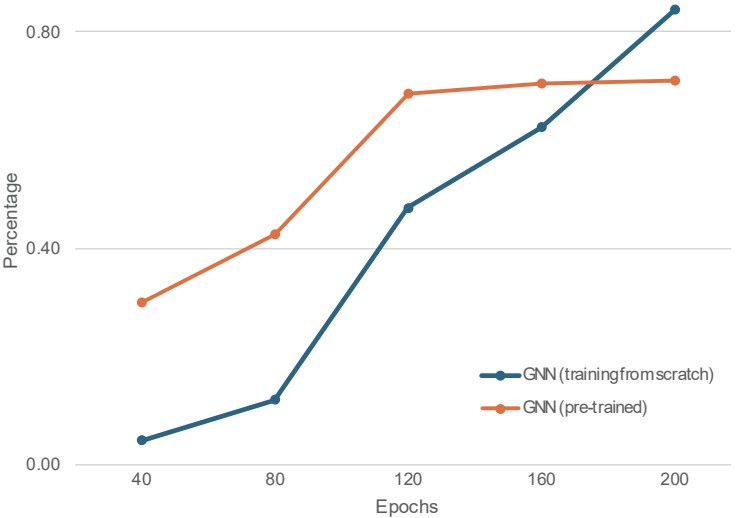

Figure 7: **Evolution of the GNN explainer during RL training.**

explainer learned from policy-update graphs collected under different task sequences in the same environment. This pretrained explainer performs worse than the explainer trained jointly with the current RL agent. A likely reason is distribution shift: different task sequences induce different policy-update distributions, while joint training allows the explainer to track the current agent's evolving policy.

