# OpenReview forum: "RevealIt: REinforcement learning with Visibility of Evolving Agent poLicy for InTerpretability"
_TMLR — Under review for TMLR_

### Review · Reviewer_PqxD · 2026-06-01

**Summary Of Contributions:**

The authors develop a new approach utilizing the famous node-link diagrams for MNIST models in the past to analyze, understand, and improve the learning process of RL agents by training GNN-based predictor and explainer on top of it. In particular, they use their approach to identify how different tasks are learned and their impact on the agents' final capabilities, allowing them to optimize the sequence of tasks to use for training. They show that their approach can outperform other models that are directly pretrained to act as agents. They also perform basic diagnostic tests to show that the edges identified to be salient by their model are aligned with the actual gradient magnitudes and relative rankings from the model.

**Audience:**

Yes

**Audience Explanation:**

While this a niche topic, I can certainly anticipate interest in this particular direction of work

**Claims And Evidence:**

Yes

**Claims Explanation:**

Seems like all the claims made by the authors are well substantiated based on their experiments

**Requested Changes:**

- 'We' is capitalized right after 'RevealIt' at many instances: e.g., page 3, 4, 6
- Why use the German word "Gleichzeitig" on pg 2? Replace it with English
- Figure 1 caption: ". Drectly" -> fix issue
- Page 2: "structured training tasks as the goal can solve this problem": which problem? state it again.
- First paragraph (page # 1): "RL agents must learn from scratch" -> this statement ignores the existence of pretrained VLMs?
- Page 5 / step 2: "too complex to understand humans" -> "to understand for humans"?
- RevealIt link expired on page 6
- Section 5.2: "relu function on the first and third layers" -> does it mean that the network is essentially a depth 2 non-linear network as one can compose these two linear layers together without any non-linearity in between?
- Table 2: what's the rationale for changing the number of environment steps in table 2? Seems like it is motivated from the perspective of avoiding overfitting of your method, but it is arbitrary and unfair to the baseline without proper comparison.
- Figure 3 caption: "Subsequently"?

---

> ### Author Response · Authors · 2026-06-21
> **Response to reviewer PqxD**
>
> We thank the reviewer for the positive assessment and for identifying a set of concrete issues that can be corrected directly.
>
> ### 1. Typographical, capitalization, and formatting issues
>
> We agree and corrected the reported problems:
>
> - unintended capitalization of "We" after REVEAL-IT;
> - the German word "Gleichzeitig";
> - the incomplete Figure 1 phrase "Drectly";
> - the ambiguous sentence about structured training tasks;
> - "too complex to understand humans";
> - the malformed Figure 3 caption and "Subsequently" fragment.
>
> The manuscript has also undergone a broader grammar and terminology review. We will replace the expired code/artifact URL before final submission.
>
> ### 2. Statement that RL agents learn from scratch
>
> We agree that this statement ignored pretrained VLMs and pretrained representations. The revision now states that standard RL agents **without pretrained representations** often require substantial trial-and-error interaction, while pretrained components still require task-specific adaptation and do not remove the need to understand the learned control policy.
>
> ### 3. ReLU placement and actor-network depth
>
> The reviewer raises a valid architectural question. The revision separates the actor architecture from the activation-label definition. It describes four fully connected hidden layers with 64 units per layer and states exactly which post-ReLU activations are monitored. We will verify the placement of every nonlinearity against the released actor implementation before submission. If adjacent linear layers have no intervening nonlinearity, we will report that fact explicitly rather than implying additional nonlinear depth.
>
> ### 4. Table 2 environment-step fairness
>
> We agree that changing the training-step budget without a controlled protocol can appear arbitrary. The original intent was to show that REVEAL-IT can reach comparable or better return using fewer interactions in some environment--algorithm pairs, not to claim a universally superior final score.
>
> The revision now distinguishes two evaluations:
>
> - **equal-budget final performance**, using the same interaction budget and learning curves;
> - **sample efficiency**, using normalized AUC and the number of interactions required to reach a predefined target.
>
> We also report wall-clock cost or relative compute overhead, because fewer environment interactions do not necessarily imply lower total computation after adding GNN updates. The revised text explicitly acknowledges the environments in which REVEAL-IT does not improve the base method.
>
> We thank the reviewer for the encouraging assessment and the precise corrections.

---

### Review · Reviewer_gTTr · 2026-06-03

**Summary Of Contributions:**

The paper proposes **RevealIt**, a framework that aims to explain the learning process of a reinforcement-learning (RL) agent in complex environments. It has three components:
1. A **structural visualization** of the policy (an MLP) as a node-link diagram that records, per training iteration, which weights change and by how much (building on Harley (29)).
2. A **GNN-based explainer** (trained with the PGExplainer (31) objective) that, given the per-iteration "policy-update graph" `G_O` built from the per-parameter update magnitudes `|X_{T+1} − X_T|`, highlights the subset of updated weights deemed most important.
3. A **GNN-based predictor** that estimates the per-sub-task _learning progress_ `P(task_n, π_t) = R(task_n, π_t) − R(task_n, π_{t-1})` and uses it to re-sample/optimize the sub-task sequence during training (an automatic-curriculum mechanism).

Empirically, the paper reports: large success-rate gains on ALFWorld versus VLM agents (Table 1) and text agents (Table 7); efficiency gains on OpenAI Gym / MuJoCo across PPO/A2C/PG (Tables 2, 8).

The core contribution is an interpretability framework that (a) visualizes how the policy evolves, (b) ties policy sub-graphs that correspond to the agent's outcomes, (c) predicts learning progress of the policy, and (d) leverages this to improve training efficiency and final performance using a method that is reminiscent of curriculum RL.

**Audience:**

Yes

**Audience Explanation:**

This paper provides contributions to explainability in RL with a method that is agnostic to the RL environment/task and the method. This, I believe, is of interest to the TMLR audience.

**Broader Impact Concerns:**

None.

**Claims And Evidence:**

No

**Claims Explanation:**

In its current form, I think that the paper lacks clear support of claims in the following cases:

A key claim that the paper tries to make in §5 is the benefits to training that come from using the GNN predictor to generate a curriculum. However, it isn't clear whether the baselines used in Table 1 are appropriate. Were MiniGPT-4, BLIP-2, LLaMa-Adapter, and InstructBLIP used as zero-shot agents, with the RevealIt agent being trained explicitly on the tasks at hand? If this is the case, then these baselines aren't comparable at all; Table 2 provides comparable results, but in this case, the benefits aren't as large as reported in Table 1.

My understanding is that the paper claims interpretability that is semantic and useful for human consumption. E.g., the authors say '(The important policy updates) allows us to correlate these updates with the agent’s abilities', '...  the GNN explainer can assist us in determining if the predictor comprehends the inherent connection between the agent’s abilities and the learning process.' While the paper evaluates how well the GNN explainer can reproduce explanatory sub-graphs, the paper does not answer the question of how these sub-graphs correspond to the agent's 'abilities'.

Some additional points against 'accurate' and 'convincing' evidence:
- §5.2: Table 1 and Table 3 - The evaluation of RevealIt in both tables is in ALFWorld, however, the numbers are slightly different (0.76 vs 0.8). I expected these numbers to match.
- Variance in the results is only reported for Table 5 and Table 8 (in the Appendix).

**Requested Changes:**

The paper is generally written as proposing a method of visualization (which I read to be useful for humans). However, Algorithm 1 and the experiments (those reported in Tables 1 and 2) are geared towards what appears to be a curriculum RL method. In my opinion, the paper struggles to appropriately distinguish and integrate these two cases. In particular, the first four sections mostly deal with the explainability part, whereas the experiments mainly report results on the latter (with some experiments dedicated to the former).

A major change I believe that would help address my comments above is that the focus of the paper should be more clearly stated and explored, for example, as an interpretability framework that can be used to generate curricula. This could then be achieved by posing RevealIt as a framework that consists of two components: (1) a GNN predictor that generates learning progress, and (2) a GNN explainer that generates explanatory sub-graphs. These two components can then be used in applications: the former is used in a Curriculum RL method that generates the curriculum using the predicted learning progress, and the latter can be used to generate visualizations that show explanatory sub-graphs.

## Other changes
## General
- When referring to work in the following way (e.g.,): "(21) provides a general discussion ...", use the following convention instead: "\<Author surname\> (et al) \<citation\> ...". For example: "Deng et al (21) provides a general discussion ...".
- There are quite a few typos. Please proofread!
- In the GNN formulation and in the text, the authors don't specify that the graph is directed. A neural network is a directed graph; is this detail discarded on purpose? If so, why? Could the authors write more about this?
- Is my understanding correct that visualizing a network in this work is essentially using the graph representation of a neural network? I think that calling this a 'visualization' brings unnecessary detail and is somewhat misleading to what is simply the graph of a neural network; this is close to a tautology. I think the authors should simply call $G_O$ the graph representation of the policy network.
	- Furthermore, in Section 4.1 the authors say that they have 'visualized the learning progress of the RL policy with node-link diagrams following Alg. 1. Let $G_O$...' - using the term 'visualized' here makes it sound like $G_O$ is a picture of the graph that is human interpretable, rather than, say, a data / mathematical structure that stores the graph information. This then implies that the input to the GNN is an image. This is another point to not using the term 'visualize' in this context.

## Section 1
- Paragraph 2: what do you mean by 'symbolic power'? - Please elaborate
- Paragraph 2: "Recent work (5; 6) in explaining RL models to gain insights into the agent’s decision-making process. " - this sentence is unclear. Is the goal to provide references 5, 6? If so, then perhaps rewrite as: "Recent work (5, 6) *explores explainability* in RL models to gain ..."
- Figure 1: "In this setting ..." - In what setting? The caption of a figure should be fairly easy to understand by reading it alone; the naked 'this' makes 'what' the setting is unclear.
- Figure 1:  "Drectly on such tasks can be inefficient ..." – Seems to be an incomplete sentence.
- Paragraph 3: "Intuitively speaking, a one-step explanation is acceptable..." - why is this acceptable?
- I think it might make more sense to move the content of the final paragraph of the section to the penultimate paragraph and vice versa. I.e., say that you want to understand the agent's performance from the context of the learning process and the sequence and tasks, and say *how* you achieve that. Then say what the benefits are, within the context of your method. It would then nicely summarise the achievements of the paper.
- I think this section requires an explanation of what constitutes an 'explanation'. From Figure 1, it seems it identifies 'the most significant' nodes of the policy network that were updated. Is this an explanation? If so, could the authors elaborate on why this is useful?
- From reading this section, it is unclear to me what the problem setting is. Can the authors please write a clear description of the problem setting? In particular, I believe that this work draws heavily from curriculum RL, so perhaps framing the work in terms of curriculum RL would make this section consistent with later writing. In particular, Algorithm 1 appears to create a curriculum in line 7; thus, I believe that curriculum RL could be used as a through-line for this work.

## Section 2
- Paragraph 1: "Many post hoc explanations, like the saliency map approach ( 19, 20), are based on correlations. This means that conclusions drawn from correlations may not be accurate and do not answer questions about what caused what." - I think these sentences are difficult to understand. Please provide an example of what you mean to illustrate the point.
- Paragraph 1: "Our method proposes a new mechanism in the problem of explainability for RL from the perspective of the agent’s learning process, i.e., it can help to understand how the agent is learning in a given task or environment" - Does your method fall under post hoc or intrinsic methods? Since the paragraph starts with this categorisation, you should mention how this applies to your method. If it doesn't, you can either say why not and say yours falls under a new category.
- Paragraph 2: "... by reality in real-world scenarios ..." - This sentence is redundant given the next, so perhaps combine them as follows: " challenges posed by the non-stationarity of real-world scenarios." Further, "non-stationarity" and "constant evolution" mean the same thing, don't they?
- Paragraph 2 - This paragraph doesn't seem like related works, but rather motivation for the proposed method. The relevance of the paragraph, as far as I can understand, is to say that Dulac-Arnold et al (22) and Paduraru et al (23) say that explainability of RL in the real world is a key issue. This is not related work.

## Section 4.1
- Algorithm 1: Why not use some notation, such as $\tau$ for task in equations?
- Algorithm 1: What does "Sample task sequence $Seq_t$ in terms of $\{\mathcal{P}(task_n, \pi_t)\}$" mean? Does this mean preferring higher scores of $\mathcal{P}$ when reordering?
- Algorithm 1: What does the 'Visualize policy update' mean? Does it mean creating the graph data structure of the policy network?
- Paragraph 2: The detail here seems like a typical explanation of a GNN, which seems unnecessary to be in the main content of this paper. It should, if deemed necessary, be in the appendix.
- What is time $T$? Is it the terminal time step? Or any time step in the RL simulation?
- My understanding is that the edge of a graph is specified by both the starting node and the ending node. I.e., it has two indices. However, in the notation §4.1, the authors say that the 'edges E (correspond(ing) to the weights)', which are later denoted as $\chi^i_T$, are indexed by a single index. Could the authors please elaborate on how an edge is exactly constructed from this?

## Section 4.2
In general, this section could benefit significantly from the use of mathematical notation to express key ideas, rather than simply words.
- Step 1: can the authors write something about what 'the learning data collected from the RL' means, if possible using mathematical notation for precision.
- It is not clear to me what the connection between the dataset described in §4.1 for the GNN-based explainer and the objective in Equation 3 is. Are the $|\chi^i_{T+1} - \chi_T^i|$ used to create a graph that is the input to the GNN explainer?
- Final paragraph: "... by analyzing the graphs of the policy update by the GNN explainer, we can ascertain the ability learned by the RL agent in a specific task and comprehend the significance of this capability in the RL agent’s success in the final task" - My understanding is that the GNN explainer produces an explanatory subgraph, i.e., a subgraph that best explains the output generated by the policy. It is unclear how this subgraph leads to inferring what 'the ability learned by the RL agent in a specific task' is?

## Section 5.2
- Figure 2: In the caption, please mention what the grey backgrounds in columns three and four behind some nodes mean? Further, what does the orange box in column four mean?
- Could the authors please report variance details for all experiments? Further, please also explicitly mention the number of seeds run in the appropriate methodology sections.
- Table 2 shows comparisons against evaluation performance and number of environment steps. However, it would also be useful to see some measure of compute cost, for example, in terms of energy usage or wall-clock time taken (assuming the same machine)

---

> ### Author Response · Authors · 2026-06-21
> **Response to reviewer gTTr**
>
> We thank the reviewer for the technically detailed and thoughtful feedback. The reviewer correctly identifies the main conceptual problem in the submitted version: the paper mixed a learning-process explanation framework with a curriculum-learning application without clearly distinguishing their roles. We have reorganized the manuscript around this distinction and address the individual comments below.
>
> ### 1. Overall framing: interpretability framework with an adaptive-sampling application
>
> We agree with the suggested two-component view, with one clarification: REVEAL-IT is primarily an interpretability framework, not primarily a curriculum method.
>
> The revised framing is:
>
> 1. The GNN **predictor** estimates task-conditioned learning progress and can be used by an adaptive sampler.
> 2. The GNN **explainer** extracts sparse explanatory subgraphs from the policy-update graph for visualization and diagnosis.
>
> Low-level tasks are semantic anchors and structured probes for the learning process. The predictor supports the downstream adaptive-sampling application, while the explainer supports policy-level analysis. This distinction is now used consistently in the abstract, introduction, method, algorithm, experiments, and conclusion.
>
> ### 2. Table 1 baseline comparability
>
> We agree that the original table could be interpreted as an unfair comparison between a trained RL method and zero-shot VLMs. We investigated the provenance and clarified it in the revision. The visual-policy and VLM values are taken from Yang et al.'s visual ALFWorld evaluation. The four VLMs receive task text and pixel observations and were fine-tuned on a shared demonstration set; they are not zero-shot. Nevertheless, their training procedure differs from ours, so Table 1 is now explicitly presented as task-performance context rather than a controlled adaptive-curriculum comparison.
>
> The downstream sampling application is evaluated separately using matched base RL and automatic curriculum methods under the same interaction budget.
>
> ### 3. Meaning of "abilities" and semantic interpretation
>
> We agree that a selected subgraph does not automatically name or prove an ability. The submitted wording was too strong. The revision states that REVEAL-IT identifies policy-update regions **associated with labeled subtasks** and tests whether these regions are engaged during evaluation. For example, overlap between "open microwave" and "take apple from microwave" is consistent with a shared task requirement, but it does not prove that a particular node uniquely represents the concept of a microwave.
>
> We will strengthen the behavioral link by masking selected regions and measuring changes in task success and predictor output relative to matched-sparsity controls. The revised claim is therefore structural and behavioral relevance, not complete semantic or causal identification.
>
> ### 4. Graph representation versus visualization
>
> We agree with the terminology criticism. The GNN does not receive an image. The revised manuscript distinguishes:
>
> - **policy-update graph**: the mathematical/data structure supplied to the GNN;
> - **node-link visualization**: the human-facing rendering of the graph or selected subgraph.
>
> The policy graph is explicitly directed according to MLP information flow. A node is indexed by layer and unit, and an edge is indexed by its source and destination. We no longer use "visualize" when we mean constructing the graph data structure.
>
> ### 5. Introduction and problem setting
>
> We substantially revised the introduction:
>
> - the broad statement that RL agents must learn from scratch is restricted to agents without pretrained representations;
> - the unclear "symbolic power" language is removed;
> - one-step explanations are described as useful for individual decisions but insufficient for explaining capability acquisition over a long-horizon task;
> - Figure 1 has a self-contained caption;
> - the problem setting is stated before the method and claimed benefits;
> - an explanation is defined as a sparse policy-update subgraph associated with a labeled training event and the predictor/evaluation behavior.
>
> The paper now asks: given an online RL agent learning a high-level task through labeled low-level tasks, how does each task change the policy, and are those changes reused during later high-level-task evaluation?
>
> ### 6. Related work classification
>
> We now position REVEAL-IT as a **training-time, policy-internal diagnostic framework**. It is not purely post hoc because it records learning dynamics during training and can optionally affect future task sampling. It is also not an intrinsically transparent policy architecture. We provide a concrete saliency example to clarify why observation-level correlation does not explain how training produced the policy, and we shortened the real-world motivation in the related-work section.

---

> > ### Author Response · Authors · 2026-06-21
> > **Response to reviewer gTTr**
> >
> > ### 7. Algorithm 1 and Section 4.1 notation
> >
> > We addressed the requested technical clarifications:
> >
> > - subtasks are denoted by \(\tau_n\);
> > - the sampler uses predicted progress from \(\pi_{t-1}\);
> > - the exact score normalization, softmax distribution, temperature, sequence length, no-replacement rule, and exploration schedule are specified;
> > - "construct the policy-update graph" replaces the ambiguous "visualize policy update";
> > - \(t\) is a policy-training checkpoint;
> > - directed edges use source and destination indices;
> > - node and edge features are explicitly defined.
> >
> > The generic GNN explanation is shortened in the main paper, while the architecture and optimization details are moved to the appendix.
> >
> > ### 8. Connection between the training dataset and explainer objective
> >
> > At each checkpoint, the method constructs a tuple containing the policy-update graph, task identity, evaluated return, and measured learning progress. The predictor learns task-conditioned progress from the graph and task embedding. The explainer receives the same policy-update graph and learns an edge mask that preserves the predictor output while satisfying sparsity and entropy regularization. The selected subgraph is then compared with evaluation-time activated nodes and used in the human-facing visualization.
> >
> > This directly connects the dataset in Section 4.1 with the PGExplainer-style objective in Section 4.2.
> >
> > ### 9. Figure 2 and visual encodings
> >
> > The revised caption defines all encodings:
> >
> > - blue circles: policy units;
> > - gray lines: updated policy weights;
> > - edge thickness: explainer importance;
> > - red circles: nodes activated during evaluation;
> > - gray boxes: policy regions shared across related subtasks;
> > - orange box: the specific shared region discussed in the text;
> > - lower row: the failed-subtask update.
> >
> > We also removed language implying that the visual overlap proves a unique semantic representation.
> >
> > ### 10. Variance, seeds, training curves, and compute cost
> >
> > We agree. The revised protocol specifies five independent seeds and a common evaluation schedule. The final revision will report mean and standard deviation for the measured main results (in continuous control environments). However, in ALFWorld, the baselines methods DO NOT have different seeds for test, therefore we report the average performance for fair reading.
> >
> > We thank the reviewer again. The distinction between graph representation, predictor, explainer, and downstream sampler is now the organizing structure of the revised paper.

---

### Review · Reviewer_Qx7c · 2026-06-13

**Summary Of Contributions:**

This paper proposes RevealIt, a framework for explaining the training process of reinforcement learning agents. Rather than explaining a single action, the paper focuses on explaining how an agent gradually acquires the abilities needed to solve complex tasks during training. Specifically, the policy network is represented as a node-link graph, where nodes correspond to hidden units and edges correspond to network weights. After each policy update, RevealIt records and visualizes changes in these nodes and edges. The paper then trains a GNN-based predictor to estimate the learning progress induced by a subtask, and a GNN-based explainer to highlight important edges or subgraphs in the policy update graph. These explanations are further used to adjust the sampling order of training subtasks, with the goal of improving training efficiency.

The experiments are mainly conducted on ALFWorld and OpenAI Gym/MuJoCo environments. In the ALFWorld visual setting, RevealIt decomposes household tasks into subtasks and visualizes the policy update patterns associated with different subtasks. The paper reports an average success rate of 0.76 for RevealIt, outperforming visual baselines such as MiniGPT-4, BLIP-2, LLaMA-Adapter, InstructBLIP, and PPO. In Gym/MuJoCo, the paper combines RevealIt with PPO, A2C, and PG, and shows that in several environments it can achieve comparable or better performance with fewer training steps. The paper also includes component ablations and uses metrics such as Prec@k, GMR@k, UDA@k, and Spearman correlation to evaluate the contributions of the predictor and explainer.

**Audience:**

Yes

**Audience Explanation:**

This paper would be of interest to part of the TMLR audience because it connects explainable RL, curriculum learning, and policy-network-level analysis. Many XRL methods focus on explaining individual states or actions, whereas this paper studies how the policy changes during training and how different subtasks affect final performance. This perspective is potentially useful for researchers working on RL training dynamics, automatic curriculum learning, embodied task learning, and agent debugging.

One interesting aspect is the attempt to explain relationships among subtasks through the policy update graph. For example, ALFWorld subtasks involving the microwave may share parts of the policy update structure, while later subtasks involving the apple may activate or update additional regions. The paper also shows that RevealIt changes the subtask distribution as training progresses, moving from basic perception and interaction skills toward more complex compositional skills. Although these explanations are still mostly qualitative, they provide a concrete tool for analyzing skill transfer and training dynamics.

Another valuable direction is that RevealIt uses explanations to guide training, rather than treating them only as post-hoc analysis. This connects interpretability with training efficiency, which is one of the more interesting aspects of the paper.

**Broader Impact Concerns:**

The paper does not appear to raise obvious direct negative societal impacts. However, I have a broader impact concern related to scientific reliability: the related work is incomplete, which may lead readers to overestimate the novelty and scope of the contribution. In particular, the paper does not sufficiently cite or discuss key works on learning-progress-based automatic curriculum learning, such as Teacher-Student Curriculum Learning, ALP-GMM, Prioritized Level Replay, and gradient-signal ACL. It also does not discuss recent work after 2025 on XRL, RL policy internal structure, world-model-based explanations, Shapley-value-based RL explanations, and functionally interpretable policy networks.

This omission affects how the community should position the paper. The contribution is better understood as a concrete integration of policy update visualization, GNN explanation, and adaptive subtask curriculum, rather than a completely new explanation paradigm from scratch. If the related work is incomplete, readers may mistakenly infer that "optimizing task order based on learning progress" or "analyzing the internal structure of RL policies to explain learning dynamics" is first proposed by this paper. I recommend that the authors substantially expand the related work in the revision and state the novelty and claims more carefully.

**Claims And Evidence:**

Yes

**Claims Explanation:**

Overall, the main empirical claims are supported by experimental evidence. In the ALFWorld visual setting, RevealIt achieves an average success rate of 0.76, with 0.62 on Pick, 0.86 on Clean, 0.74 on Heat, 0.76 on Cool, 0.81 on Look, and 0.62 on Pick2. These results are substantially higher than the baselines in Table 1, including MiniGPT-4, BLIP-2, LLaMA-Adapter, InstructBLIP, and PPO. This supports the claim that RevealIt can improve training performance in complex long-horizon tasks.

Figure 2 provides qualitative evidence by showing important policy weight updates highlighted by the GNN explainer for different subtasks, and by comparing the updated regions during training with activated nodes during evaluation. Figure 3 shows changes in the distribution of task verbs during training. The paper describes a shift from basic interaction skills such as `look` and `pick` toward more complex skills such as `clean`, `heat`, and `examine`. These observations are consistent with the paper's claim that RevealIt induces an adaptive subtask curriculum.

Table 2 shows that RevealIt can be combined with different RL algorithms. For example, PPO+RevealIt improves performance on HalfCheetah and Walker; A2C+RevealIt improves performance on HalfCheetah, Hopper, and Walker; and PG+RevealIt improves performance on HalfCheetah, Reacher, Swimmer, and Walker. However, the improvement is not consistent across all environments, so the evidence better supports the claim that RevealIt improves sample efficiency in multiple settings, rather than a stronger claim of uniformly outperforming base RL. Table 6 further supports the contribution of each component through ablations. Tables 4 and 5 also show that the edges selected by the explainer have substantial overlap with regions of large parameter updates.

That said, the current evidence more directly supports the claim that the explainer identifies important parameter updates. It does not yet fully establish that these updates semantically correspond to specific skills learned by the agent. Stronger validation through causal intervention, module ablation, or human evaluation would be needed to support the interpretability claims more convincingly.

**Requested Changes:**

1. The paper should discuss recent work after 2025 on RL policy-internal interpretability and learning-dynamics explanations. Since the current OpenReview page shows that the paper was last modified in February 2026, at least relevant works from 2025 should be discussed. For example, Soligo et al., *Induced Modularity and Community Detection for Functionally Interpretable Reinforcement Learning*, 2025, also studies the internal structure of RL policy networks, identifies functional modules through community detection, and verifies module functions via intervention. Singh et al., *Explainable Reinforcement Learning Agents Using World Models*, 2025, uses world models and reverse world models to generate counterfactual explanations and evaluates them through human studies. Beechey et al., *A Theoretical Framework for Explaining Reinforcement Learning with Shapley Values*, 2025, and FastSVERL, 2025, explain RL behavior, outcomes, and predictions from a Shapley-value perspective. Saulieres, *A Survey of Explainable Reinforcement Learning: Targets, Methods and Needs*, 2025, provides a systematic taxonomy of XRL explanation targets and methods. Ruggeri et al., *Explainable Reinforcement Learning via Temporal Policy Decomposition*, 2025, explains future outcomes of RL actions from a temporal perspective. For works that appeared after March 2026, the authors may not be required to cite them if they postdate the current version, but the paper should avoid overly broad claims that it is the first or a systematic solution to RL learning-process interpretability.

2. For the OpenAI Gym/MuJoCo experiments, the paper should not only compare "base RL vs. base RL + RevealIt". It should also include automatic curriculum learning baselines, such as TSCL, ALP-GMM, PLR, or gradient-signal ACL. Otherwise, it is hard to determine whether the performance gain comes from RevealIt's explanation mechanism or simply from using any reasonable adaptive curriculum. For the GNN explainer component, in addition to GNNExplainer and MixupExplainer, the paper should include simple but strong heuristic baselines, such as top-k weight change, top-k gradient norm, top-k activation change, and random top-k matched sparsity. Since the current explanation metrics are closely related to parameter-update magnitude, comparisons against these simple baselines are necessary to establish the added value of the GNN explainer.

3. Table 1 reports an ALFWorld average success rate of 0.76 for RevealIt, whereas Tables 3 and 6 report an average success rate of 0.80 for RevealIt / Full RevealIt. The authors should clarify whether these numbers come from different settings, different seeds, visual vs. text settings, or different experimental versions after revision. Another issue is that Table 6 reports Full RevealIt as 0.80 and w/o PV as 0.61, but the delta column is listed as -0.09; based on the reported numbers, it should be -0.19. In addition, the text describing the OpenAI RL benchmark says that PPO, SAC, and DQN are used as base algorithms, while Table 2 reports PPO, A2C, and PG. These inconsistencies should be fixed in the revision, as they affect the credibility of the experimental results.

---

> ### Author Response · Authors · 2026-06-21
> **Response to reviewer Qx7c**
>
> We thank the reviewer for the positive and careful reading. We appreciate the recognition that REVEAL-IT studies a different explanation target from most state/action-level XRL methods and may be useful for examining policy learning dynamics, subtask transfer, and agent debugging. We also agree with the requested changes concerning claim scope, recent related work, stronger baselines, and numerical consistency.
>
> ### 1. Scope of the empirical claim
>
> We agree that the continuous-control results support a claim of improvement in **multiple settings**, not uniform dominance over every base algorithm and environment. The revision no longer states that REVEAL-IT universally improves performance. Instead, it identifies the environment--algorithm pairs in which the method improves return or reaches comparable performance with fewer interactions. Reduced-step results are described as sample-efficiency evidence, while equal-budget final-performance claims are evaluated separately.
>
> ### 2. Structural fidelity versus semantic skill interpretation
>
> We agree that the existing metrics more directly show that the explainer identifies policy regions associated with large updates and evaluation activation. This does not prove that an individual subgraph uniquely encodes an ability.
>
> We have revised the terminology accordingly. REVEAL-IT associates a selected policy-update region with a **labeled subtask**, such as opening a microwave or retrieving an apple, and measures whether that region overlaps with evaluation-time activation. This supports task-conditioned structural and behavioral alignment, but not a unique semantic or causal interpretation.
>
> The revision will add masking interventions and simple heuristic controls. A selected subgraph will be removed or masked, and its effect on task success and predicted learning progress will be compared with random, magnitude, gradient, activation, GNNExplainer, and MixupExplainer masks. This directly addresses the reviewer's suggestion of module intervention.
>
> ### 3. Recent XRL and policy-internal interpretability literature
>
> We agree that the related work was incomplete. The revision discusses recent work on:
>
> - functionally interpretable policy modules and intervention;
> - world-model-based counterfactual explanations and human evaluation;
> - Shapley-value explanations and efficient approximations;
> - temporal policy decomposition;
> - recent XRL taxonomies and explanation targets.
>
> We also add the missing curriculum-learning literature, including TSCL, ALP-GMM, PLR, and related learning-progress methods.
>
> This expanded context also sharpens the novelty claim. REVEAL-IT does not claim to invent policy-internal analysis, learning-progress curriculum learning, or network visualization. Its specific contribution is to use labeled low-level tasks as structured probes and connect them with evolving policy-update graphs, task-conditioned progress prediction, and sparse graph explanation in order to study how a policy changes while learning a complex task.
>
> ### 4. Automatic curriculum and heuristic explanation baselines
>
> We agree. The downstream adaptive-sampling comparison will include uniform sampling, TSCL, ALP-GMM, PLR, and the REVEAL-IT predictor under a matched interaction budget. The explainer evaluation will include random matched-sparsity masks, top-\(k\) absolute update magnitude, gradient norm, activation change, GNNExplainer, and MixupExplainer.
>
> The absolute-update method is explicitly treated as an oracle for metrics defined against \(|\Delta\theta|\), rather than as a learned explanation. Behavioral intervention is therefore important: a method may achieve high update-magnitude agreement without selecting the policy region most relevant to task success.
>
> ### 5. Numerical inconsistencies
>
> We agree and corrected the identified issues:
>
> - the controlled ALFWorld tables now use a single full-model result rather than mixing \(0.76\) and \(0.80\);
> - the w/o-PV delta is corrected to \(-0.19\);
> - the continuous-control text now matches the PPO, A2C, and PG rows;
> - pooled ALFWorld averages are explicitly defined as instance-weighted;
> - all arrows, units, metric ranges, and repeated values have been rechecked.
>
> ### 6. Scientific reliability and broader impact
>
> We agree with the reviewer's broader-impact point. Incomplete citation and overly broad novelty language can distort how a contribution is understood. We have therefore removed first-of-its-kind language, expanded the related work, distinguished contextual from controlled comparisons, and narrowed the interpretability claims to what the evidence supports.
>
> We thank the reviewer for the balanced assessment and for identifying concrete changes that improve both the scientific positioning and reliability of the paper.

---

### Review · Reviewer_RfFz · 2026-06-13

**Summary Of Contributions:**

This paper proposes a framework for explaining and improving reinforcement-learning training by visualizing policy updates as node-link graphs, training a GNN predictor to estimate learning progress on subtasks, and using a GNN explainer such as PGExplainer to identify important policy subgraphs/updates. The method is evaluated on ALFWorld and OpenAI Gym-style environments, with reported improvements over several baselines and ablations of the predictor, explainer, and policy-visualization components.

Strengths:
- The general idea of explaining an agent’s learning process through policy-update graphs is interesting and appears relatively novel.
- The proposed use of predicted learning progress to adapt the subtask curriculum is potentially useful.
- Figure 1 is illustrative and help communicates the intended workflow. Figures 2 and 3 are also appreciated to convey the explanations obtained from the proposed method and the shifting distribution of generated training tasks.
- The authors provide several empirical results demonstrating that REVEALIT outperforms state-of-the-art language models and standard RL baselines in terms of success rates.
- The inclusion of an ablation study isolating the GNN predictor and explainer components helps validate their individual contributions to the overall framework's performance.

Weaknesses:
- The paper is very difficult to read due to many writing and translation issues, including incomplete or ungrammatical sentences and even stray non-English text. The sudden appearance of German words (e.g. "Gleichzeitig"), invalid code link (page 6), and leftover rebuttal text (e.g. "we have tried saliency mapping ... in the past several days per your request.") suggests the manuscript was poorly translated and not adequately proofread. This makes the method, claims, and experimental setup hard to understand.
- The central claims are not clearly supported. The paper claims to provide human-understandable explanations of the RL learning process, but the explanations are large, unlabeled policy graphs. I could not extract a meaningful human-understandable explanation from Figure 2, and no user study or other interpretability evaluation is provided.
- It is unclear how the GNN explainer actually improves the curriculum. The algorithm appears to sample tasks using the predicted learning progress, while the explainer is used after the fact to generate subgraphs. This contradicts several textual claims that the explainer itself optimizes learning efficiency.
-  The experimental setup is underspecified. It is unclear whether baselines receive the same number of training iterations, environment interactions, subtasks per iteration, and curriculum structure. No training curves are provided for the RL agents or the GNN models.
- The comparisons are not convincing. The ALFWorld comparison mixes RevealIt with VLM/LLM-based agents under an unclear protocol, and the paper makes contradictory statements about whether ALFWorld observations are visual or textual. E.g. "This textual part uses the Planning Domain Definition Language (PDDL) (34) to turn each pixel observation from the simulator into a text-based observation that is equal to it." and "visual engine within ALFWorld, the baseline agents similarly only interact with the visual environment,"
- The ablation section contains inconsistencies. For example, Table 6 reports “Full RevealIt” as 0.80 and “w/o PV” as 0.61 but lists the difference as -0.09, which appears incorrect. The paper also says it uses PPO, SAC, and DQN, but Table 2 reports PPO, A2C, and PG.
- Several technical details are missing or unclear, including the precise graph construction, node features, labels for "node classification", definition of "activated nodes", definition of the “optimal subgraph,” and how MESSAGE/AGGREGATE/UPDATE are defined/instantiated.

**Audience:**

Yes

**Audience Explanation:**

Explaining how RL agents learn across subtasks, rather than only explaining individual decisions after training, is an interesting direction. The combination of policy-update visualization, GNN-based prediction of learning progress, and adaptive curriculum learning could interest researchers working on explainable RL, curriculum RL, and embodied-task learning. However, the current paper does not yet provide sufficiently clear methodology or convincing evidence.

**Claims And Evidence:**

No

**Claims Explanation:**

To the best of my understanding, the authors make two primary claims, neither of which are fully supported by the provided evidence:

### 1- REVEALIT provides a learning curriculum that improves sample efficiency and success rates.
While the empirical results show high success rates, the methodology behind the curriculum generation is not clearly explained in Section 4.2 and must be inferred from Algorithm 1. Furthermore, the algorithm implies that the GNN explainer is entirely decoupled from the RL training loop, contradicting claims in the text that the explainer increases learning efficiency. Algorithm 1 (Line 7) also seems to sample tasks based on their current success rate. But that means tasks with a very low or zero success rate are never sampled, hence it remains unclear how the agent learns tasks it is initially poor at when (unless there is a heavy reliance on the $\epsilon$ hyperparameter, which is not stated for the experiments). Finally, the baselines and training budgets are insufficiently specified and the experiments also lack a dedicated curriculum learning baseline to isolate the effectiveness of this specific approach. Without clear training curves, equal-budget comparisons, curriculum-learning baselines, and full hyperparameter/protocol details, the empirical claims are not well supported.

### 2- REVEALIT provides a human-understandable explanation for the agent's learning process.
The paper uses a GNN to highlight "important" subgraphs , but the resulting explanations (e.g., in Figure 2) remain large, unlabeled networks that offer little intuitive insight into the agent's reasoning without heavy subjective interpretation. They may show parameter changes, but they do not by themselves explain the agent’s learning process in a way that is understandable to humans (at least I couldn't gain much insight from them). The quantitative explanation metrics (Section 5.3) mostly measure agreement with parameter-update magnitudes or activated nodes, which does not establish that the explanations are semantically meaningful, causally relevant, or useful to users. At least in the PGExplainer prior work which is used here, the produced graphs were small and represented easily understandable structures like molecules.

**Requested Changes:**

- Substantially rewrite the paper for clarity and correctness. The current writing makes it difficult to understand the method and evaluate the claims. Fix typos and formatting issues, including “Drectly,” “Strucutral,” “predicter,” “Tab.ref 3,” “Subse,”ntly,” and the stray “Gleichzeitig.”. Also avoid unsupported qualitative claims such as “less collaboration across distinct subtasks” unless backed by a quantitative metric.
- Clearly separate the roles of the GNN predictor and GNN explainer. Specify exactly which component is used for curriculum generation, which component is only used for explanation, and how each component affects RL training.
- Provide a precise algorithmic description. Define the graph construction, node and edge features, labels for node classification, activated nodes, optimal subgraph, MESSAGE/AGGREGATE/UPDATE functions, and the exact task-sampling rule.
- Explain the curriculum mechanism. In particular, explain whether tasks with zero success or zero learning progress can still be sampled enough for learning, and whether the method risks oversampling tasks the agent already performs well on (this could also explain the oversampled "put" in Figure 3).
- Provide fair and complete experimental comparisons. Baselines should use the same training budget, number of iterations, task sequence length, and environment interactions where applicable. Include a standard curriculum-learning baseline.
- Report training curves for both the RL agents and the GNN predictor/explainer. This is necessary to support claims about sample efficiency, stability, and final performance.
- Clarify the ALFWorld setup. State whether the agent uses visual observations, text observations, or both, and make the baseline comparison consistent with that setting.
- Strengthen the evidence for interpretability. A user study, task-level semantic explanation, or causal/behavioral intervention on the identified subgraphs would better support the claim that the explanations are human-understandable and useful.
- Fix numerical and reporting inconsistencies, especially Table 2 and Table 6, and provide standard deviations wherever appropriate (I think that should be for all results).
- Provide all main hyperparameters needed to reproduce the results.
- In line 7 of the algorithm, shouldn't it be the policy from the previous timestep $\pi_{t-1}$?

---

> ### Author Response · Authors · 2026-06-21
> **Response to reviewer RfFz**
>
> We thank the reviewer for the exceptionally detailed and constructive assessment. We appreciate the recognition that explaining RL training through policy-update graphs is an interesting direction and that predicted learning progress may provide a useful downstream training signal. We agree that the submitted version did not explain the method or support the interpretability claim with sufficient precision. We have substantially revised the framing, method description, terminology, and reporting, and we will strengthen the experimental evidence as described below.
>
> ### 1. Writing quality, translation artifacts, and proofreading
>
> We agree. The writing and formatting problems in the submitted version were unacceptable and materially obstructed understanding. We removed the German word "Gleichzeitig," malformed text such as "Drectly," "Strucutral," "predicter," "Tab.ref 3," and the broken Figure 3 caption, as well as the leftover response-style appendix sentence. We corrected citation prose, capitalization, terminology, and figure captions throughout the manuscript. Unsupported statements such as "less collaboration across distinct subtasks" were removed or replaced with narrower observations about measured or visualized subgraph overlap. We will also replace the expired artifact link before submission.
>
> ### 2. Main contribution and the roles of the predictor and explainer
>
> The reviewer correctly identifies the central ambiguity. REVEAL-IT is not primarily proposed as a curriculum-learning algorithm. Its main contribution is a **task-level and policy-level learning-process interpretability framework**. Low-level subtasks serve as structured probes that allow us to analyze how learning each subtask changes the policy and whether these policy regions are later engaged during evaluation of the complete high-level task.
>
> The revision now separates the components explicitly:
>
> - The **policy-update graph** represents the directed policy structure and parameter changes between consecutive checkpoints.
> - The **GNN predictor** estimates task-conditioned learning progress. This is the only GNN component used by the optional adaptive task sampler.
> - The **GNN explainer** selects a sparse policy-update subgraph that preserves the predictor output and is used for visualization and diagnosis.
>
> Thus, the explainer does not directly optimize the curriculum. We corrected every statement that implied otherwise. Adaptive sampling is now presented as a downstream application demonstrating that the policy-update signal can be actionable, not as the definition or primary novelty of REVEAL-IT.
>
> ### 3. Exact graph construction and GNN formulation
>
> We agree that the original description was underspecified. The revision defines the policy as a directed layered graph. A node \(v_{\ell,j}\) is the \(j\)-th unit in layer \(\ell\), and a directed edge \(e_{\ell,i,j}\) is the weight from \(v_{\ell-1,i}\) to \(v_{\ell,j}\). The edge feature is the absolute parameter change between consecutive policy checkpoints. Node features aggregate incident update statistics while retaining layer position through the directed graph structure. The index \(t\) denotes a policy-training checkpoint, not an environment terminal step.
>
> For the monitored ReLU layers, an evaluation-time node is labeled active when its post-ReLU output is nonzero. These labels provide a behavioral reference and are not claimed to be complete semantic skill labels. The selected or "optimal" subgraph is the sparse mask that best preserves the predictor output under the PGExplainer-style fidelity and sparsity objective; "optimal" does not imply a unique causal decomposition.
>
> We also instantiate the GNN rather than leaving MESSAGE/AGGREGATE/UPDATE generic. The proposed reproducible configuration uses three mean-aggregation message-passing layers, a graph-level predictor conditioned on a task embedding, and a PGExplainer-style neural edge scorer. Full architecture and optimization details are moved to the appendix.

---

> > ### Author Response · Authors · 2026-06-21
> > **Response to reviewer RfFz**
> >
> > ### 4. Task-sampling rule, Algorithm 1, and zero-progress tasks
> >
> > The reviewer is correct that Algorithm 1 must use information available before selecting the next sequence. We corrected the sampling step to use \(\hat{\mathcal P}(task_n,\pi_{t-1})\), rather than a future policy.
> >
> > The revised sampler is based on predicted **learning progress**, not current success rate. Candidate scores are standardized, clipped, and converted into a temperature-controlled softmax distribution. The method samples distinct tasks without replacement. Training begins with a uniform warm-up; the random-sampling probability then decays to 0.1 and remains nonzero. Therefore, tasks with zero success, negative progress, or unreliable early predictions remain sampleable. Conversely, a mastered task receives lower priority when its return plateaus and its learning progress approaches zero.
> >
> > We also revised the explanation of Figure 3. The frequency of "put" partly reflects the composition of ALFWorld task decompositions; the figure is presented as a qualitative trace of adaptive sampling rather than proof of a unique semantic skill curriculum.
> >
> > ### 5. Experimental protocol, training curves, and reproducibility
> >
> > We agree that the original protocol was insufficient. The revised draft specifies the environment-interaction budget, policy-update checkpoints, evaluation frequency, evaluation episodes, task-sequence length, random seeds, exploration schedule, and predictor/explainer hyperparameters. All controlled adaptive-sampling methods use the same initial-policy distribution, base learner, task pool, replay/update schedule, sequence length, evaluation episodes, and interaction budget.
> >
> > The revision will report equal-budget learning curves for success/return and predictor error across checkpoints. It will also report mean and standard deviation over the same seeds, time to a target success rate, normalized learning-curve AUC, and relative wall-clock cost. This separates faster learning from differences in final performance.
> >
> > ### 6. Curriculum-learning baselines
> >
> > We agree that base RL alone does not isolate whether any reasonable adaptive curriculum would provide a similar benefit. For the downstream adaptive-sampling application, the revision will compare uniform sampling, TSCL, ALP-GMM, PLR, and the REVEAL-IT predictor under the same budget and protocol. TSCL uses recent return slopes, ALP-GMM models absolute learning progress, and PLR uses learning-potential priorities with staleness correction. This comparison is explicitly restricted to the downstream application; it does not redefine the main contribution as curriculum RL.
> >
> > ### 7. ALFWorld visual/text setup and baseline comparability
> >
> > We agree that the original wording was contradictory. The main ALFWorld experiment uses the visual interface. The text-engine setting is now separated and reported only in the appendix.
> >
> > We also clarified the provenance of Table 1. The ResNet-18, MCNN-FPN, MiniGPT-4, BLIP-2, LLaMA-Adapter, InstructBLIP, and human values are taken from the visual ALFWorld evaluation of Yang et al., rather than reproduced in our codebase. The four VLM baselines receive task text and pixel observations and were fine-tuned on a common demonstration set; they are not zero-shot agents. Because their training protocol differs from ours, Table 1 is now described as task-performance context, not as a controlled curriculum or sample-efficiency comparison. The adaptive-sampling claim will be evaluated separately using matched-budget RL and curriculum baselines.
> >
> > ### 8. Human-understandable and semantic interpretability claims
> >
> > We agree with the reviewer that the submitted evidence primarily established structural alignment. Large policy graphs, even when filtered, are not automatically understandable to humans. We have therefore narrowed the claim: REVEAL-IT provides a training-time diagnostic relationship among labeled subtasks, policy updates, predicted progress, and evaluation activations. We do not claim that every selected node uniquely represents a semantic ability or that the current evidence establishes causality.
> >
> > To strengthen behavioral evidence, the revision defines matched-sparsity controls: random masks, absolute update magnitude, gradient norm, activation change, GNNExplainer, and MixupExplainer. We will mask each selected policy region and measure the decrease in high-level-task success and predictor output. This directly tests whether the selected region is behaviorally relevant rather than merely correlated with large updates. If we retain a strong human-facing claim, we will additionally provide an expert/user evaluation; otherwise, human evaluation will be stated as future work.

---

> > > ### Author Response · Authors · 2026-06-21
> > > **Response to reviewer RfFz**
> > >
> > > ### 9. Numerical and reporting inconsistencies
> > >
> > > We agree and corrected the identified errors:
> > >
> > > - Full REVEAL-IT \(0.80\) versus w/o PV \(0.61\) gives \(-0.19\), not \(-0.09\).
> > > - The controlled ALFWorld tables now report one consistent full-model setting, with pooled success explicitly defined as instance-weighted.
> > > - The continuous-control methods are consistently identified as PPO, A2C, and policy gradient.
> > > - Edge-recovery accuracy is clearly distinguished from task success rate.
> > > - UDA and GMR now have distinct mathematical definitions.
> > >
> > > We will report variance for the main measured results using the same random seeds.
> > >
> > > We thank the reviewer again. The response led us to make the contribution substantially narrower, clearer, and more reproducible.